# Investigating the '*Bolsonaro effect*' on the spread of the Covid-19 pandemic: An empirical analysis of observational data in Brazil

**Mireille Razafindrakoto**[1]*, **François Roubaud**[1], **Marta Reis Castilho**[2], **Valeria Pero**[2], **João Saboia**[2]

**1** Institut de Recherche pour le Développement, LEDa-DIAL Research Unit, IRD, Université Paris-Dauphine, PSL Research University, Paris, France, **2** Institute of Economics, Federal University of Rio de Janeiro (UFRJ), Rio de Janeiro, Brazil

* mireille.razafindrakoto@ird.fr

**Data Availability Statement:** The data underlying the results presented in the study are available from: Ministry of Health, Secretarias Estaduais de Saúde. Brasil. Regarding Covid-19, data are

## Abstract

Brazil counts among the countries the hardest hit by the Covid-19 pandemic. A great deal has been said about the negative role played by President Bolsonaro's denialism, but relatively few studies have attempted to measure precisely what impact it actually had on the pandemic. Our paper conducts econometric estimates based on observational data at municipal level to quantitatively assess the '*Bolsonaro effect*' over time from March 2020 to December 2022. To our knowledge, this paper presents the most comprehensive investigation of Bolsonaro's influence in the spread of the pandemic from two angles: considering Covid-19 mortality and two key transmission mitigation channels (social distancing and vaccination); and exploring the full pandemic cycle (2020–2022) and its dynamics over time. Controlling for a rich set of relevant variables, our results find a strong and persistent '*Bolsonaro effect*' on the death rate: municipalities that were more pro-Bolsonaro recorded significantly more fatalities. Furthermore, evidence suggests that the president's attitude and decisions negatively influenced the population's behaviour. Firstly, pro-Bolsonaro municipalities presented a lower level of compliance with social distancing measures. Secondly, vaccination was relatively less widespread in places more in favour of the former president. Finally, our analysis points to longer-lasting and damaging repercussions. Regression results are consistent with the hypothesis that the '*Bolsonaro effect*' impacted not only on Covid-19 vaccination, but has affected vaccination campaigns in general thereby jeopardizing the historical success of the National Immunization Program in Brazil.

## 1. Introduction

The Covid-19 pandemic hit Brazil shortly after the Carnival in 2020. Following a first case reported in São Paulo on 25 February, the pandemic swept through state capitals. Measures to check the spread of the virus were adopted in mid-March, but the number of daily cases had

provided at https://data.brasil.io/dataset/covid19/_meta/list.html; Mobility data are available at https://data.humdata.org/dataset/movement-range-maps?fbclid=IwAR1c-FwWACCVdbmOd1qX2jpsX4EWM2oCL_SIPYKtrDq5an5TvXADNgMiwNg; Vaccination data are available at https://opendatasus.saude.gov.br/dataset/covid-19-vacinacao/resource/301983f2-aa50-4977-8fec-cfab0806cb0b Data on election are available at https://sig.tse.jus.br/ords/dwapr/r/seai/sig-eleicao-resultados/resultado-da-elei%C3%A7%C3%A3o?session=117201719502.

**Funding:** This research received no external funding.

already topped the thousand mark by the end of the month. Within a few months, Brazil had become one of the hardest hit countries in the world, setting a tragic record that still stands today. By the end of 2022, Brazil had the second-highest absolute number of Covid-19 deaths behind the United States, with official sources reporting almost 700,000 deaths. The negative impact of President Bolsonaro's decisions and attitude on the mortality rate became a focus of attention right from the start. He was one of the three presidents of note singled out (along with Donald Trump and Lopez Obrador) for their attitudes and statements described as irresponsible in the latest report by the Lancet Commission on Covid-19 [1].

The denialist stance taken by the president, at the head of the federal government, was marked by his rejection of scientific evidence and especially his disputing the effectiveness of lockdown measures against the pandemic [2]. President Bolsonaro's first response was to downplay the gravity of the virus, referring to the disease as a 'little flu' and discouraging social distancing and other measures adopted by subnational (state and municipality) governments [3]. His main argument was that the economic implications would be worse than the health impacts. The Brazilian federal government's actions to combat the pandemic suffered from a lack of coherent policies and coordination among the different government entities [4]. This resulted in delays and disorganization in the implementation of measures. Sub-national legislative bodies and the judiciary (Supreme Court) have played a positive role in trying to neutralise Bolsonaro's policies [5]. Yet it also caused confusion among the population: some people failed to understand or challenged the measures adopted by municipalities and states. An analysis of Bolsonaro's 'denialism' points out that the subnational governments ended up taking the lead in fighting the pandemic crisis, albeit not without presidential resistance to their actions and initiatives [2, 6]. Both governors and opposition parties have used the judicialization of politics to deal with the Covid health crisis. In fact, the president loses more than he wins in the Federal Supreme Court on issues related to the management of the pandemic–especially in the judgment of the constitutionality of Provisional Measures (MPs), changing the trend that usually tends to favour the federal government [5, 7].

Therefore, alongside the different characteristics usually considered to explain the incidence of Covid-19 (comorbidities, age, colour/race, housing or working conditions, etc.), political factors may have an impact on national performances in handling the pandemic. In concrete terms, the mechanism at play, noted mainly for the United States and Brazil, associates the population's behavioural response to the pandemic with government officials' rhetoric and action. These factors have an influence on the people's perception of risk and thereby impact on the extent of compliance with pandemic mitigation measures. Previous studies have already shown that political leaders' actions and narratives can affect support for public policies and individual behaviour in representative democracies. In the case of health policies in particular, few studies prior to the outbreak of Covid-19 pointed up a link between certain political regimes and long-term health indicators such as infant mortality or the vaccination [8, 9]. This literature joins for obvious reasons political science studies which seek to understand the influence and mechanisms used by certain political groups in the implementation of public policies. In the case of the pandemic, several studies show that populist governors were at the head of countries with the worst levels of pandemic incidence [10–12].

In the USA, a number of studies have analysed the relationship between counties' political profiles and their respective populations' attitudes towards the pandemic. A study [13] shows that interest in the pandemic and compliance with mobility restrictions were lower in *counties* where Trump won in the 2016 presidential election. They find that the way a given message is interpreted can vary depending on the information source and/or the political affiliation of the person providing the information. Other studies on the USA confirm the differences in behaviour and risk perception by ideological or political orientation (See, for example, [14–16]). In

the case of Brazil, various academic papers in political sciences and public health point up Bolsonaro's responsibility in the catastrophic handling of the pandemic. An analysis [17], comparing Bolsonaro, Dutertre and Trump's responses to the outbreak of the pandemic, forges the concept of 'medical populism' characterized by the following features: simplifying the pandemic by downplaying its impacts or touting easy solutions or treatments, spectacularizing their crisis responses, forging divisions between the 'people' and dangerous 'others', and making medical knowledge claims to support the above. Two analysts [18] argue that the lack of public health governance can best be described as governance without (central) government based on 'strategic ignorance'. The role of social media, misinformation and fake news is also key on this topic [19–22] (for a more general discussion at the global level, [23]). From a general point of view, many articles blame presidential right-wing populism in Brazil [24] and other countries for the catastrophic handling of the pandemic [25–30]. A typology of 'populist' response types [12], covering 29 parties and leaders in 22 countries around the world, classifies Bolsonaro as a 'COVID radical'. As stressed by one of these studies [30], Bolsonarism can be considered the most expressive recent experience of radical right-wing populism in an emerging country, that led to a health and economic disaster in the management of the pandemic.

In this context, our paper aims to assess the quantitative effect of President Bolsonaro's behaviour on the development of the pandemic in Brazil, which we call the 'Bolsonaro effect' in keeping with former studies [31, 32]. In one of the pioneering analyses of the influence of the president's behaviour on the evolution of the pandemic in Brazil [33], the authors perform an econometric analysis of the Covid-19 contamination rate in municipalities where Bolsonaro won more than 50% of the total votes in the first round of the 2018 election. They use a difference-in-differences approach to compare the situation in municipalities before and after the pro-Bolsonaro demonstrations of 15 March 2020. The authors show that the municipalities where the demonstrations occurred recorded more hospitalizations and deaths than the others. The authors believe that this effect is due both to people crowding into the demonstrations and to 'laxer' attitudes to social distancing in keeping with the president's rhetoric and position.

The extent to which the president's rhetoric at the start of the pandemic in 2020 impacted social distancing has been assessed. One study [34] shows that people's mobility in pro-Bolsonaro municipalities (identified by the 2018 election results) consistently increased in the week following the president's actions and speeches to downplay the impacts of the pandemic and discourage compliance with social distancing. This effect was stronger in municipalities with a significant local media presence, a large number of Twitter accounts and a high proportion of evangelicals (a significant base of Bolsonaro's support). Similar results are obtained by other studies [35, 36].

Further papers, looking beyond the earliest stage of the pandemic, confirm the existence of a 'Bolsonaro effect' [37–40]. By way of example, one study shows that the correlation at the States level between Bolsonaro's support and fatalities is increasing over time. They estimate that 57% of the total number of deaths registered at the end of 2021 are due to political factor [40]. In line with the national and international literature on the influence of the political dimension in the evolution of the pandemic, considering a wide range of determinants of the incidence of and number of deaths from Covid-19 the 'Bolsonaro effect' is one of the factors that proves to be robust in explaining the pandemic [31, 32]. Interestingly, one study seeks to go further by introducing a placebo test for other respiratory diseases and disentangling the 'Bolsonaro effect' from traditional right-wing ideological orientations. In this setting, the 'Bolsonaro effect' persists, at least at the early stage of the pandemic [30]. Finally, one of the rare studies which take a population level approach, based on an individual survey conducted nationwide in December 2020, shows that Bolsonaro's supporters were less likely to consider

the pandemic as a key challenge, less worried about getting infected and less likely to wear face masks [28].

Building on the abovementioned studies, this article aims to underline the far-reaching impact of the president's denialist stance in Brazil. It provides added value to the literature by consolidating and deepening existing studies on the repercussions of Bolsonaro's statements and actions on the spread of the pandemic in the country from three points of view. Firstly, there is the question of the persistence of the effects over time. A major effort has therefore been made to cover the entire period of the pandemic using stabilized and harmonized data. We assess the extent to which explanatory factors changed or not from March 2020 to December 2022. Secondly, we shed light on the main mechanisms by which political factors ultimately affected Covid-19 mortality rates. For this purpose, we systematically consider the '*Bolsonaro effect*' on the two main indicators related to the mitigation measures: the mobility rate, which quantifies the effectiveness of social distancing policies (defined here as staying at home and away from others as much as possible); and the vaccination rate, which evaluates the level of compliance with pharmacological recommendations. Thirdly, a wide range of robustness tests, including analytical extensions, are used to ensure the relevance of our approach to identify and measure the '*Bolsonaro effect*'. First, different proxy variables are considered to check the reliability of data and indicators, while contributing to the discussion. Second, we set out to disentangle the '*Bolsonaro effect*' from the effect of right-wing voter behaviour and anti-science movements, since the latter effects would have occurred without Bolsonaro's involvement.

To sum up, three conclusions are drawn from the analysis. First, our results support the view that the impact of the president's denialist stance persisted over time. The effect appears to be more pronounced during Covid-19 waves when the contamination rate surged. Second, we provide evidence that the "*Bolsonaro effect*" affected mortality and behaviour in terms of social distancing and vaccination. Therefore, impact channels at least in part through the two mitigation measures. Firstly, pro-Bolsonaro municipalities posted a lower level of compliance with social distancing measures. Secondly, vaccination was relatively less widespread in places more in favour of the former president. Delving deeper into these results, it turns out that the Bolsonarists' anti-vax narrative had a stronger effect on young people and whites in pro-Bolsonaro municipalities. Third, not only do the different tests support our results, but they also underline the extent of the impact. Brazil faces longer-lasting damaging repercussions beyond the effect of Bolsonaro's rhetoric on Covid-19 vaccinations. Regression results are consistent with the assumption that the '*Bolsonaro effect*' has impacted vaccination campaigns in general (not just the Covid-19 campaign), thereby jeopardizing the success of the National Immunization Program in Brazil.

Following this introduction, Section 2 presents the Brazilian context: the development of the pandemic and the main measures adopted. The third section presents the databases and empirical strategy. Section 4 discusses the results and performs multiple robustness checks and extensions. The last section concludes with some final considerations.

## 2. The Brazilian context

### 2.1 Covid-19 incidence, mobility and vaccination

Brazil is among the countries with the highest numbers of Covid-19 deaths, whether in absolute terms or relative to its population with official sources reporting nearly 700,000 deaths by the end of December 2022, the country ranks second after the United States (Table 1). Brazil still leads the board in relative terms, although Peru and a few Eastern European countries have higher death rates. This macabre record stands even when considering the notorious

**Table 1. The five countries reporting the most Covid-19 deaths worldwide.**

|  | Official data (31/12/2022) | | Excess mortality (Estimates as at 31/12/2021) | | | |
|  |  | | Wang *et al.* (2022) | | WHO (2022) | |
|  | Nb deaths (in 1,000) | Mortality rate (per million) | Nb deaths (in 1,000) | Mortality rate (per million) | Nb deaths (in 1,000) | Mortality rate (per million) |
|---|---|---|---|---|---|---|
| USA | 1,093 | 3,231 | 1,130 | 3,340 | 932 | 2,755 |
| Brazil | 694 | 3,223 | 792 | 3,678 | 694 | 3,223 |
| India | 531 | 375 | 4,070 | 2,874 | 4,741 | 3,348 |
| Russia | 386 | 2,666 | 1,070 | 7,390 | 1,072 | 7,404 |
| Mexico | 331 | 2,597 | 798 | 6,261 | 626 | 4,912 |
| World | 6,693 | 839 | 18,210 | 2,283 | 14,910 | 1,869 |

*Sources*: Our World in Data (https://ourworldindata.org/), Wang *et al.* (2022) and WHO (2022) [41, 42].

underestimation of official data. Two independent exercises based on excess mortality estimates confirm the Brazilian tragedy [41, 42]. At the end of 2021, Brazil ranked 4th and 5th worldwide due to the upward re-estimation of mortality in India, Russia and Mexico. It is worth mentioning the relatively good quality of Brazil's official data [43] due in particular to the historical strength of its public health system and a team effort by the media, which took over when Bolsonaro's government decided to stop publishing the daily death count (see below).

The pandemic struck in three waves (Fig 1). It first spread rapidly, reaching an initial level of approximately 1,000 deaths per day between May and August 2020. The number of deaths then fell slowly, but started rising again in October 2020 with the appearance of new variants. At the height of the second wave, the country was recording more than 3,000 deaths per day (March 2021). The ramping up of the vaccination programme saw a sharp drop in the number of deaths through to December 2021. The arrival of the Omicron variant at the start of 2022 brought a third wave of contamination and death, but one that was less intense and shorter than before. Since then, the number of deaths has remained at a low level, albeit still averaging more than 100 deaths per day in December 2022.

The pandemic's dynamics can be equated with the development of two of the main measures taken to contain it, namely social distancing and vaccination. Data from *Google Mobility Reports* reveal a sharp and massive drop in mobility in Brazil in March and April 2020 (Fig 1). This was the result of lockdown policies adopted at different levels (federal, state and municipal), their extent of adoption and enforcement, and individual decisions to stay at home. The population's mobility on the whole ran parallel to the mortality dynamics. The peak of the first wave, for example, was followed by a gradual return to normal. By the end of 2020, individual mobility had returned to its pre-pandemic level. The second wave saw another reduction in mobility, but on a smaller scale than during the first wave due to the difficulty (observed in most countries) of imposing restrictions similar to those adopted in the early months of the pandemic. As of the end of 2021, Brazilians started to move around unhindered again, with the exception of a slight downturn in January 2022 when the third wave struck.

The vaccination campaign got off to a slow start due to resistance from and negligence on the part of Bolsonaro's government, but took off properly in the second quarter of 2021. Leveraging the experience gained by the National Immunization Program, 60% of the population had received a first dose by August 2021, representing nearly 2 million people vaccinated per day (Fig 1). The height of the campaign appears to have been reached in mid-2022 when more than 85% of first doses had been administered and nearly 80% of Brazilians had completed a full Covid-19 vaccination schedule. Since then, rates have only increased marginally.

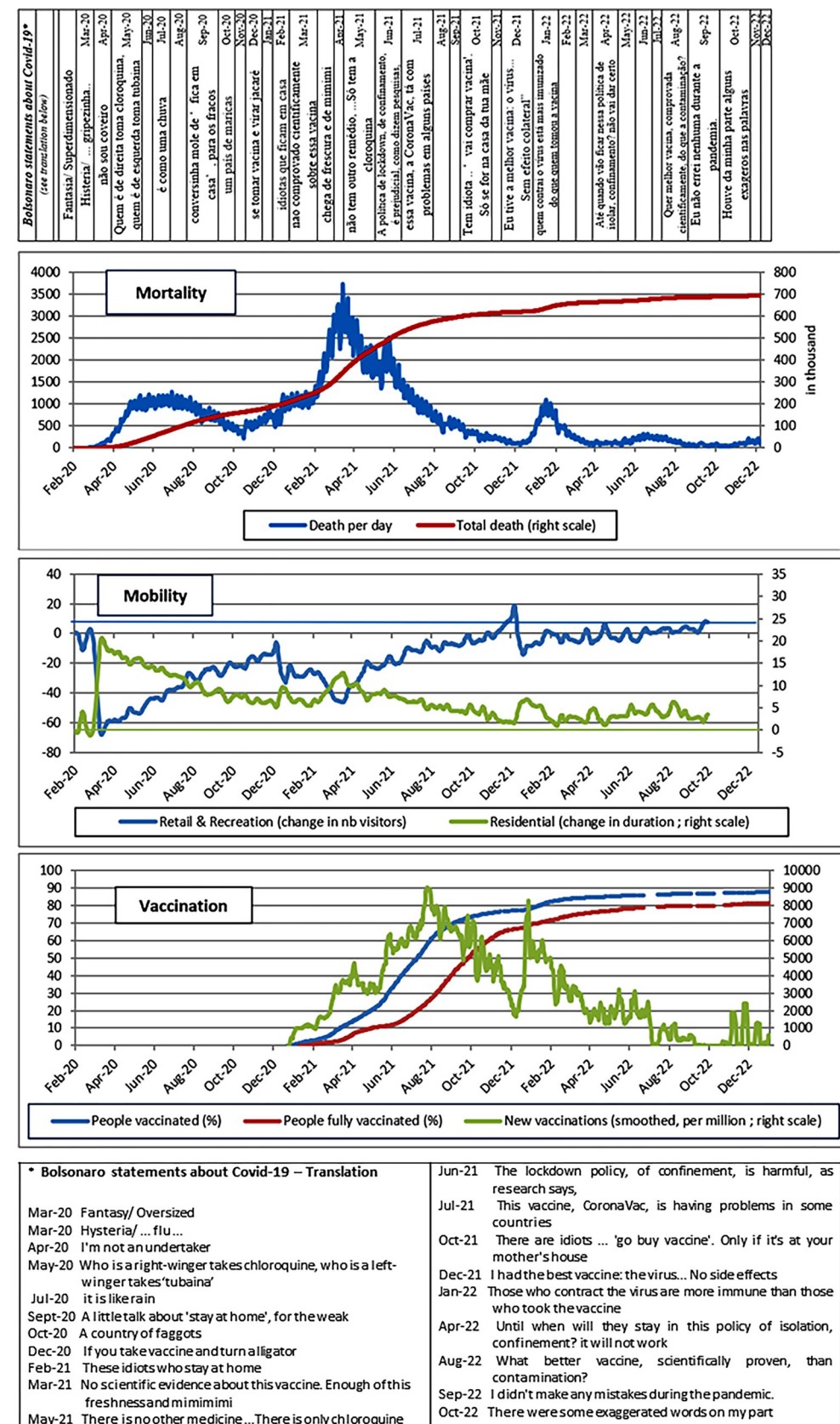

**Fig 1. Growth in mortality, mobility and vaccination rates in Brazil.** Sources: Ministry of Health, Secretarias Estaduais de Saúde. Brasil. For the mobility data: Google mobility report (change relative to January 2020), for Bolsonaro's statements: https://www.aosfatos.org/, Authors' own elaboration. Note: Data considered are the seven-day moving average for each variable.

## 2.2 Public policy in response to the pandemic

Brazil's main particularity in its handling of the pandemic was President Bolsonaro's attitude and speeches designed to downplay it, if not deny its existence. This default (anti) health policy used the full panoply of imaginable methods and rhetoric: denial, conspiracy, disputing scientific evidence, fake news, etc. The President consistently denied the gravity of the pandemic and the effectiveness of measures, pharmaceutical or otherwise, when the vaccines were developed. Fig 1 and S1 Table contain some illustrative examples of Bolsonaro's vocal denialist statements over the period.

The responsibility for pandemic response policy measures in Brazil lies with different government levels–municipal, state and federal. Effective action calls for a combination of the visions of these three government levels and naturally the coordination of actions taken to ensure their complementarity and coherence.

Yet social distancing measures were not only adopted in an uncoordinated manner without federal government support, but they were also the target of attacks and criticism by the president (S1 Table). He even challenged them in court and encouraged events and demonstrations to be held against them [34]. The actions of the two sub-national government levels were often contradictory: in March 2020, all states closed their schools and at least certain non-essential businesses, but the economy was reopened in accordance with local plans (at municipal level) with different measures and timelines. The absence of an adequate testing and screening policy further undermined the performance of local plans and the success of the non-pharmacological interventions [44, 45].

The Ministry of Health handled non-pharmacological measures differently depending on the minister in charge of the portfolio–four different ministers headed up the Ministry of Health over the period observed. Whereas the first health minister to tackle the pandemic (Luiz Mandetta) tried to coordinate the actions of the different government bodies (hospitalization, social distancing, personal protective equipment, etc.), the minister in charge of the portfolio for the longest time (Colonel Pazuello) did not encourage non-pharmacological measures such as social distancing and face masks. In keeping with the President of the Republic's rhetoric, he refrained from launching awareness campaigns and promoted the use of medicines and treatments deemed inefficacious by scientists.

In terms of vaccination, the government's actions mirrored the problems found on other fronts in the fight against Covid-19 with negative campaigns, a lack of coordination and transparency, and defiance of science (S1 Table). Vaccine procurement came late and in short supply, and the vaccination issue itself triggered controversies and disputes between the federal government and governors and mayors of different regions. Some epidemiologists put this forward as one of the explanatory factors for the second wave. Yet despite the late start and repeated vaccine supply problems (as shown by the low vaccination rate in July 2021; Fig 1), the vaccination rate increased relatively quickly in Brazil due to the prior existence of a universal primary healthcare system (SUS) and a well-established immunization programme [46]. The National Immunization Program (NIP) set up in 1973 is classed as one of the most successful immunization programs in the world, with a remarkable vaccination capacity in terms of geographical coverage and agility, and a track record of rolling back other epidemics [2]. Despite the coverage of the SUS and its resilience to low investment in recent years [44, 47],

pandemic handling conditions differed widely across regions. The lack of centralized coordination of actions and resource distribution meant that the pandemic spread differently across regions. Spatial disparities in income and resources across Brazilian regions, as well as between town and country, are well known and can be seen from the imbalanced supply of medical resources nationwide [43, 48].

In view of the federal government's position and suspected flaws in the handling of the Covid-19 pandemic, a Parliamentary Committee of Inquiry (CPI) was set up in April 2021 to investigate the federal government's responsibility in the spread of the pandemic in the country and, in particular, the worsening health crisis in Amazonas. In October 2021, the CPI delivered its report. Bolsonaro was "*proven to be primarily responsible for the mistakes made by the federal government during the Covid-19 pandemic*" [49]. He was found guilty of nine charges including prevarication, charlatanism and, above all, crimes against humanity.

In terms of economic policy, the measures taken were both classic and paradoxical: classic, because they were similar to those taken by many other countries; and paradoxical, because a massive programme of emergency transfers to households was put in place by an ultra-liberal government that had always opposed redistribution policies [3]. The federal government's actions were based primarily on two sets of emergency measures to address the negative effects of the pandemic [50]: i) fiscal measures to offset household income losses, support businesses and provide financial assistance to states and municipalities; and ii) liquidity support and regulatory capital measures to ensure the stability of the financial system and expand the supply of credit [51, 52].

Emergency Aid (*Auxilio Emergencial*, AE) was the main income guarantee mechanism for informal workers in vulnerable situations. Launched in April 2020, the amount provided in the first three months corresponded to about 60% of the minimum wage and was three times higher than the iconic conditional cash transfer programme *Bolsa Familia*. Furthermore, the programme was rolled out on an unprecedented scale, providing aid more than 67 million Brazilians or nearly one-third of the population. The benefit was halved in the last four months of 2020. In 2021, the programme's renewal came up against strong resistance within the government itself. It ended up being renewed from April to October for a lower amount and fewer beneficiaries.

The Brazilian government also launched an income guarantee programme for formal workers in the shape of Emergency Benefit for the Preservation of Employment and Income (BEM). Under this programme, the federal government supplemented workers' wages pro rata to their reduction in working hours. The benefit was extended until December 2020 and was relaunched in April 2021 when the second wave of the pandemic struck.

The emergency package was estimated at more than 10% of GDP [52], the largest programme in Latin America, and similar in size to those implemented in developed countries. The two flagship measures, the BEM and especially the AE, were successful in protecting the income of low-income workers. According to an estimation [53], the income of poor households increased slightly in 2020 (0.2%), despite the loss of almost 8 million mainly informal jobs, compared with a drop of 4.2% for middle-income households and 1.2% for the wealthiest.

## 3. Materials and methods

### 3.1 Empirical approach

The empirical analysis is based on three estimation models. The first, which addresses the main question of this study, estimates the correlation between the political factor (the '*Bolsonaro effect*') and the Covid-19 mortality rate. The other estimation models explore two possible

transmission channels through which the political factor could have affected the mortality rate: the mobility rate and the vaccination rate are indeed found to impact on the number of Covid-19 deaths.

The first dependent variable (Eq 1) is the cumulative Covid-19 mortality rate (cumulated at date t or for a time period from $t_1$ to $t_2$). The second dependent variable (Eq 2) is an indicator of mobility that measures the extent to which people decided for themselves to stay at home or social distance or complied with stay-at-home or social distancing policies/recommendations. The variable considered is the ratio of people who left home compared to a reference period (February 2020). The third dependent variable (Eq 3) is the vaccination rate (the ratio of vaccinated people to the municipality's population). Two indicators are used: the percentage of people who received at least one dose of a Covid-19 vaccine (vaccinated 1 dose), and the percentage of people fully vaccinated (those who completed the primary vaccination series (2 doses for most vaccines). The estimated models consider a broad spectrum of controls (S2 Table) to cover the maximum number of types of variables with a potential direct or indirect effect on the mortality rate, and above all, confounding variables which potentially might have influence on the mortality rate and the political factor.

The unit of analysis is the municipality. Information was collected and structured from various sources for the 5,570 municipalities in Brazil. This is the smallest administrative entity for which comprehensive data on Covid-19, as well as political and sociodemographic characteristics, are available for the whole country. Moreover, it captures collective (or neighbourhood) and individual behaviour effects. Another reason for this choice is that many policies are designed and implemented at municipal level. In the case of Covid-19, the lack of central government coordination placed the onus for pharmacological and non-pharmacological measures more on the states and municipalities, giving rise to different policy responses with various repercussions on the death rate. For all these reasons, the municipal approach is relevant and advantageous.

This type of approach has its limitations and the results should be interpreted with due caution. First, analysis by municipalities cannot be interpreted in terms of individual risks. However, we know that a significant effect at municipal level also guarantees significance in terms of individual probabilities. As a matter of fact, If the mortality rate is different between two population categories (e.g. Poor and Non-Poor) at aggregate level, it should also be different at individual level (likewise, if the mortality/contamination rate is the same at individual level, it should also be the same at aggregate level). We can assume, then, that the individual and municipal approaches generally converge in terms of signs [54]. Otherwise, inverse mechanisms would need to be explained. For example, if the municipalities with higher male ratios are harder hit, it does not necessarily mean that males are hardest hit. Women in municipalities where male ratios are high could be most at risk of infection. However, this would imply finding a particular mechanism to support this view.

Second, the econometric models tested here can be used to estimate the correlations between the mortality rate and different factors, corrected for structural effects. However, as in most analyses of observational data, we identify correlations that do not necessarily imply causality. For example, it is quite plausible that restrictions were applied more strictly in municipalities where mortality rates were already higher. It is therefore hard to disentangle the actual impact of the restrictive measures. Finally, there is the issue of potentially omitted variables correlated with political factors. In this case, the correlation observed in our model might be influenced by these omitted variables. Yet it is hard to find convincing arguments to support this reservation. Considering the data on comorbidities most often cited as examples of omitted variables, it is not clear to what extent they might be correlated with political factors.

Nevertheless, although we do not identify any actual causal relationships, our multiple regression framework considering information on all Brazilian municipalities can give policy-makers a better understanding as to whether or not there is a conditional correlation between political factors and the Covid-19 mortality rate. This at least helps rule out certain hypotheses about potential causal mechanisms.

## 3.2 Mortality rate and political factors

It is worth mentioning that the mortality rate was preferred over the incidence rate (number of confirmed cases) due to possible problems resulting from low levels of testing and underreporting of cases. According to an estimate [43], Covid-19 case underreporting at as high as 70% in the first semester of 2020. Thus, the mortality rate is modelled to identify the characteristics of municipalities affected by the pandemic and changes in patterns from the start of the pandemic to the end of 2022. Given the non-normality of the data on the number of deaths and the over-dispersion of count data, the parameters are estimated by a negative binomial model using the maximum likelihood method, an approach already used in similar study [55]. This model was also estimated under an OLS specification for similar results and the same order of magnitude.

The model is applied to 12 quarters which correspond to the different phases of the pandemic (see Fig 2).

The specification of the Covid-19 mortality rate model is as follows:

$$Y_i = exp(a_0 + \alpha B_i + \beta C_i + u_i) \tag{1}$$

Where:

$Y_i$ represents the dependent variable—mortality rate (per 100,000 inhabitants in municipality i).

$B_i$ is the main variable of interest related to the '*Bolsonaro effect*'.

$C_i$ is the vector of the control variables (sociodemographic characteristics alongside some direct transmission factors at municipal level associated with characteristics and place of residence).

$u_i$ is the error term.

The main variable of interest in terms of political factors is the percentage of votes for President Bolsonaro per municipality in the first round of the 2018 election. We chose the 1st round instead of the 2nd because we capture the core of Bolsonaro's supporters. The purpose of the percentage of voters is to capture what we call the '*Bolsonaro effect*', which reflects the

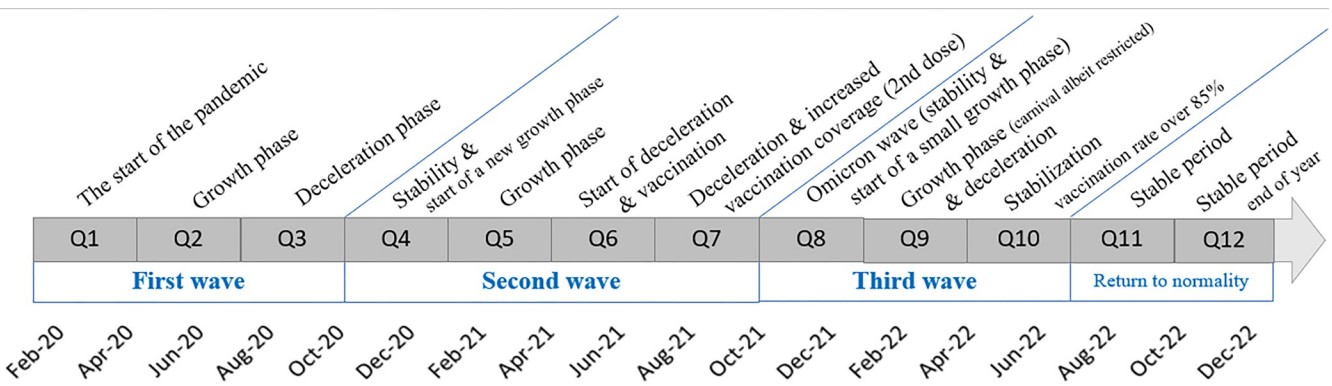

**Fig 2. Covid-19 pandemic timeline in Brazil.** Source: Ministry of Health, Secretarias Estaduais de Saúde. Brasil, Authors' own elaboration.

proportion of the population that aligned with the president's denialist position. This population ended up taking stances with harmful collective health effects by not wearing face masks or not respecting social distancing measures [28]. They also hindered or disrupted the adoption of more restrictive measures by local governments on which, as explained before, the onus had fallen for the implementation of policies to combat the pandemic. Their behaviour may have increased their risk of being contaminated and contaminating others.

Vector $C_i$ contains control variables relating to individual demographic and health characteristics (gender, age, race, education and health status represented by life expectancy) and socioeconomic characteristics (poverty rate measured by the ratio of Emergency Aid (EA) beneficiaries to the population, and GDP per capita). The control variables also include dwelling characteristics and household location factors that may have accelerated or reduced the transmission of the disease in the municipality: geographic variables (population density, urban/rural location, slum residence and number of residents per room) and a health infrastructure variable (number of doctors per 100,000 inhabitants). Other health infrastructure variables were considered (such as the number of hospitals and the number of beds per ICU), but the number of doctors per inhabitant proved to be the most significant factor. In addition to these classic variables, we also included as potential factors of virus dissemination a commuting and migration indicator as well as the informal employment rate (which generally implies more direct contact between producers and consumers).

Among the control variables, the vaccination rate and mobility indicator are two transmission factors that could be considered. However, their inclusion in the model would potentially absorb at least part of the effect of our variable of interest (the political factor). This could lead to overcontrol, as our interest is in the total effect [56]. In addition, it should be noted that the introduction of these two variables could engender a reverse causality problem in model 1. Vaccination started in municipalities with higher mortality rates, as in the case of Manaus. The mortality rate would also have influenced the decision to stay at home. Therefore, as we have no way of controlling for reverse causality for the mobility variable, we exclude it from model 1. In the case of the vaccination rate, when considering the 2nd dose of the vaccine, controlling for the 1st dose is a way to purge the reverse causality effect from the ratio of mortality to the number of vaccines. So, the vaccination rate remains in one of our specifications, but since it could be classified as a 'bad control' or a collider, our preferred specification also excludes the vaccination rate.

### 3.3 Two transmission mechanisms: Mobility and vaccination rate

We then devised an empirical strategy to relate the '*Bolsonaro effect*' to two key transmission mechanisms: (i) the extent of population lockdown (or mobility), which reflects the adoption of and compliance with non-pharmacological measures, and (ii) the vaccination rate, a pharmacological measure to control the disease. The purpose here was to check whether the effects observed for Covid-19 mortality rates in the previous mortality estimates were also observed for these two variables. The results suggest–at least in part–that non-compliance with stay-at-home measures and/or differences in vaccination rates per municipality could explain the number of Covid-19 deaths.

Considering people's mobility, the parameters were estimated using the ordinary least squares (OLS) method for the equation specified below.

$$M_i = a_0 + \alpha B_i + \beta C_i + u_i \tag{2}$$

$M_i$ represents the average mobility rate per municipality per month compared with February

2020 (before the pandemic). The independent variables were defined similarly to model (1) and estimates were made for the same periods of analysis.

Finally, we considered the vaccination rate as a dependent variable in order to measure the '*Bolsonaro effect*' factors. To this end, we estimated a negative binomial model for the determinants of the vaccination rate. Model (3) regresses the vaccination rate of the 1st and 2nd dose by municipality ($V_i$) on the same explanatory variables as in the previous models:

$$V_i = \exp\left(a_0 + \alpha B_i + \beta C_i + u_i\right) \tag{3}$$

Given that very few people were concerned by the start of vaccination in January 2021, we ran model (3) for the quarters from February 2021 for the first dose and from May 2021 for the second dose of the vaccine. It is worth mentioning that the number of observations in the first period considered for vaccination analysis is very small.

### 3.4 The data

Our purpose was to cover a wide range of indicators in this study, thoroughly and painstakingly processing and checking the data to ensure their reliability and relevance. The data used come from various sources: demographic census, survey data, administrative records and *big data*. The definition of the variables of analysis and data sources can be found in S2 Table. The data on Covid-19 deaths are drawn from the Ministry of Health's multi-institutional programme. We used data from *Departamento de Informática do Sistema Único de Saúde* (*Datasus)* available up to 2020 in addition to 2021 and 2022 data from the Ministry of Health Secretariat of Health Surveillance's SIM (*Sistema de Informações sobre Mortalidade*) to calculate excess mortality. The data on total vaccination coverage (other than Covid-19 vaccines) come from *Datasus* and more specifically from the National Immunization Program Information System (SI-PNI). For the Covid-19 vaccination rate, different *Datasus* files containing individual data on vaccination in the 27 states were used to obtain the most recent indicators detailed by population categories (age group, gender and race/colour). Therefore, organizing, harmonizing and concatenating the variables involved processing rough data containing tens of millions of observations. This was also the case for the poverty rate, proxied by the number of beneficiaries of the *Auxilio Emergencial* emergency aid for poor families (managed by the federal bank *Caixa*). Further work was required to match data from sources using different geographic codes for a given municipality. For example, the TSE (*Tribunal Superior Eleitoral*) covering data on presidential election results (2014, 2018 and 2022) and Facebook, which manages the *Movement Range Maps* data that we used for the mobility indicator, have their own municipality identification codes different to the IBGE. Finally, it should be noted that detailed data on mobility in Brazil are unavailable as of the end of 2021. However, as seen above, from the start of 2022, population movements returned to their normal pre-crisis level on the whole.

Most of the independent sociodemographic variables are derived from the 2010 population census (PC2010). The main advantages of this source are: a) its comprehensiveness (it provides measurements without sampling errors), and b) the wealth of information collected. In addition to the usual sociodemographic variables found in censuses (sex, age, education and migration), an earned-income module was included in the questionnaire for a representative sample of all households, accounting for around one-tenth of the country's entire population [57]. We therefore processed the micro-data on this one-tenth sample with more than 20 million individual observations. However, the main weakness of the PC2010 is that it is not up to date: some municipalities' characteristics may have changed since 2010. Nevertheless, robustness tests using different strategies show that this problem is very limited [31]. Moreover, the socio-

economic indicators have been adjusted/updated by information from the recent Labour Force Survey (PNAD Continua), which provides state-level indicators on three types of area for each state: metropolitan, urban and rural.

Last but not least among the different sources and data is average life expectancy per municipality: a synthetic indicator of the population's state of health taken from the FIRJAN municipal development index. This allows for robustness tests by providing alternative measures in terms of health, education and employment/income. More control variables (number of tests, number of days from the start of the pandemic in each municipality, etc.), are not included here as they do not significantly improve the model's performance or accuracy.

## 4. Results and discussion

### 4.1 Mortality

Table 2 presents the results of model (1) for the Covid-19 mortality rate for the entire period from February 2020 to December 2022 (cumulative data). Focusing on the '*Bolsonaro effect*', our rich set of controls is introduced progressively by block of variables in five different specifications: the unconditional correlation (col 2) and then inclusion of sociodemographic characteristics (col 3), transmission factors (col 4) and vaccination (col 5). In order to address

**Table 2. Factors associated with the Covid-19 mortality rate (cumulative data: Feb 2020 –Dec 2022).**

|  | (1) | (2) | (3) | (4) | (5) | (6) Poisson Fixed effect |
|---|---|---|---|---|---|---|
| **Vote for Bolsonaro** | **1.637****** | **1.187****** | **0.855****** | **0.918****** | **0.910****** | **0.624****** |
| **(1st round 2018)** | **(0.000)** | **(0.000)** | **(0.000)** | **(0.000)** | **(0.000)** | **(0.000)** |
| *Group of control variables* |  |  |  |  |  |  |
| Demographic and Socioeconomic (1) |  | Yes | Yes | Yes | Yes | Yes |
| Dwelling and infrastructure (2) |  |  | Yes | Yes | Yes | Yes |
| Vaccine rate (1st dose) |  |  |  | 0.00497**** | 0.00630**** | 0.00402** |
|  |  |  |  | (0.000) | (0.000) | (0.025) |
| Vaccine rate (2nd dose) |  |  |  |  | -0.00143 | -0.000640 |
|  |  |  |  |  | (0.363) | (0.735) |
| Constant | 4.865**** | -8.170**** | -10.20**** | -7.771**** | -7.826**** |  |
|  | (0.000) | (0.000) | (0.000) | (0.000) | (0.000) |  |
| Lnalpha | -1.496**** | -1.725**** | -1.797**** | -1.820**** | -1.820**** |  |
|  | (0.000) | (0.000) | (0.000) | (0.000) | (0.000) |  |
| N | 5568 | 5340 | 5269 | 5269 | 5269 | 5268 |
| pseudo $R^2$ | 0.027 | 0.042 | 0.048 | 0.050 | 0.050 |  |
| AIC | 67996.4 | 64088.9 | 62828.5 | 62709.1 | 62710.3 | 208860.5 |

*Sources*: Ministry of Health, IBGE, TSE; authors' calculations.

* $p < 0.10$

** $p < 0.05$

*** $p < 0.01$

**** $p < 0.001$

*Note*: Negative Binomial (NB) model except for the last column (Poisson model with State fixed effect).

Demographic and Socioeconomic (1) control variables are: poverty level (% of emergency transfer beneficiaries in the municipality), age (log), race (white), sex (male), education (higher), GDP per capita (log), life expectancy (log); Dwelling and infrastructure (2) control variables are: number of doctors per 100,000 inhabitants, municipality density (log), area (rural), migration (% of migrants), Job commuting (% of commuters), dwelling (overcrowding: average number of people per room), location (favela), informal job (% informal workers).

potential bias associated with unobserved, time-invariant state heterogeneity, and since the use of the Negative Binomial model with fixed effects is much debated and not recommended, the last column presents the results of the Poisson regression model with fixed effects at state level (col 6). The detailed table is available as S3 Table. Bear in mind that our main objective is to better identify the '*Bolsonaro effect*' rather than to interpret all the controls *per se*, which could be difficult for some variables given the presence of a high level of multicolinearity.

First, the '*Bolsonaro effect*' is positive, highly significant and robust across all specifications. Municipalities with a larger proportion of Bolsonaro voters consistently return a greater probability of higher Covid-19 mortality rates throughout the period. The estimated coefficient decreases slightly with the introduction of controls, from 1.6 (no controls) to 0.9 (all controls), meaning that Bolsonaro voter characteristics are associated with a higher mortality rate 'on average'. To give an order of magnitude, a 10% increase corresponds to around 24 additional deaths for 100,000 inhabitants. At national level, 10% more Bolsonaro supporters would have caused 51,000 more deaths by the end of 2022.

Second, most of our control variables are significant. Considering socioeconomic factors, the higher a municipality's poverty rate, the harder it was hit by Covid-19 mortality. This result is consistent with a growing number of studies in other countries [58]. At the same time, wealthier municipalities posted higher Covid-19 mortality rates. In fact, the epidemic started in big cities and spread faster in these cities due to the intensity of social interactions (exchanges, population movements, and diversity of economic and social activities). Hence, just as the poverty effect holds when controlling for GDP per capita, so too our results suggest that the greater the inequalities in a municipality, the higher the mortality rate.

The Poisson regression model with fixed effects at state level (last column) gives results that are very consistent with those of the negative binomial model. We find the '*Bolsonaro effect*' with the expected positive sign. However, we consider the results of the negative binomial model to be preferable. On the one hand, the lower coefficient could be due to the fact that the fixed effect captures part of the '*Bolsonaro effect*' at municipal level. As it was already stressed, the president's denialism spread to other levels of government, generally states whose governors were politically aligned with the president [6]. On the other hand, and more importantly, the data on mortality are exhibiting substantial overdispersion, such that the variance is substantially higher than the mean. When overdispersion is present in the data, fitting a standard Poisson regression model may lead to incorrect inference and predictions.

Third, introducing the vaccination rate into the equation, we find a positive and significant coefficient for the first dose, suggesting that municipalities with higher rates of vaccination had higher mortality rates. This reflects the fact that more people were vaccinated where it was most needed. However, moving on from our control for the first dose (and the potential reverse causality effect), the full vaccination schedule is found to be negatively correlated with the mortality rate, but not statistically significant. As already stressed above, the introduction of the vaccination rates as control variables could be questioned [56]. However, this step allows for a sensitivity test. And we can notice that the sign and the values of the coefficients of the different variables do not change significantly. Interestingly, the coefficient for the second dose is only significant when the '*Bolsonaro effect*' is excluded, suggesting that residents of pro-Bolsonaro municipalities are less vaccinated (see below for confirmation).

In order to consolidate our results to go beyond the static picture provided by the cumulative data, we investigate the '*Bolsonaro effect*' further by exploring the temporal patterns of the pandemic. We estimate model (1) by periods, in two settings: cumulative and by time periods. Fig 3 presents the coefficients for the most conservative equation (col.3), without vaccination. The '*Bolsonaro effect*' persists over time with positive and significant coefficients for nearly all periods and both settings. The only exceptions are the first period of the pandemic (February-

## Cumulative | By time periods

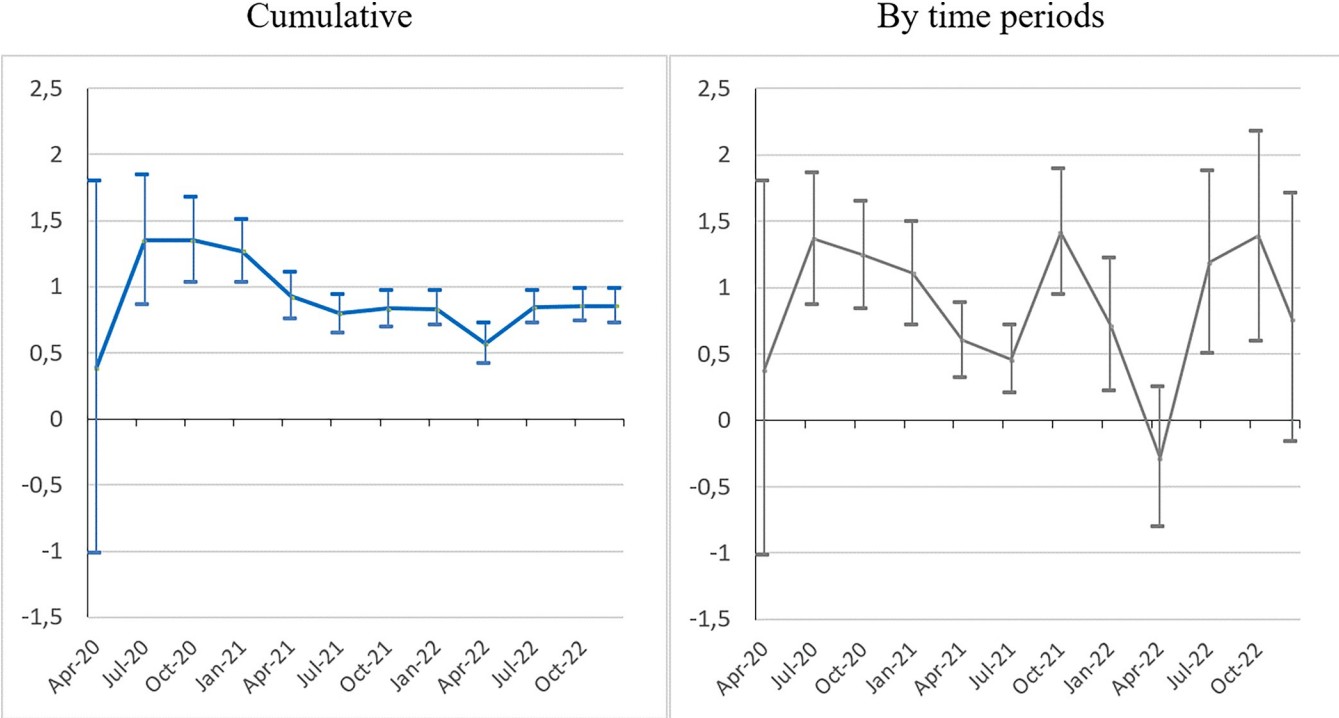

**Fig 3. The '*Bolsonaro effect*' on the mortality rate over time (2020–2022).** *Sources*: Ministry of Health, IBGE, TSE; authors' calculations. *Note*: Model (1) (see Table 2, col.3). Confidence interval at $p < 0.05$.

April 2020), still with few fatalities, and between February and April 2022, following the third wave (Omicron). By construction, the coefficient profile is smoother in the cumulative setting with the '*Bolsonaro effect*' at its height during the first wave (through to the end of 2020), and then stabilizing at a lower level through to December 2022. However, the most striking figure is the persistence of the '*Bolsonaro effect*' period after period (meaning on independent death samples) through to the end, which was far from obvious. For instance, one might have expected a 'catch-up' process whereby, after affecting pro-Bolsonaro municipalities, the pandemic would have spread to other municipalities (saturation effect). The '*Bolsonaro effect*' is the most constant over time.

As previously mentioned, voter affinity with the president's rhetoric can influence the mortality rate in several ways. His voters' low level of observance of the social distancing measures (investigated below) had individual and collective effects on the mortality rate. In addition, it had dissuasive effects on local governors in charge of implementing the social distancing measures. A survey of the implementation of non-pharmacological measures in the 27 states shows that governors' performances differed in terms of speed of response and restrictive measures depending on their political alignment with the president [6]. Our results converge with and expand on those found by other studies using quasi-experimental approaches [33, 34] conducted in the early stages of the pandemic. Moreover, they indicate that this effect persists despite observed changes in other determinants of Covid-19 mortality over time, a result in line with other analyses [32, 37, 38].

### 4.2 Change in mobility

What are the channels through which the '*Bolsonaro effect*' impacted on Covid-19 mortality? The first we can test is the main non-pharmaceutical measure (along with face mask use),

**Table 3. Factors associated with the change in mobility (cumulative data: Feb 2020 –Oct 2021).**

| | (1) | (2) | (3) | (4) | (5) | (6) | (7) |
|---|---|---|---|---|---|---|---|
| **Vote for Bolsonaro** | -0.0510**** | 0.0613**** | 0.0607**** | 0.0585**** | 0.0586**** | 0.0573**** | |
| **(1st round 2018)** | (0.000) | (0.000) | (0.000) | (0.000) | (0.000) | (0.000) | |
| *Group of control variables* | | | | | | | |
| Demographic and Socioeconomic (1) | | Yes | Yes | Yes | Yes | Yes | Yes |
| Dwelling and infrastructure (2) | | | Yes | Yes | Yes | Yes | Yes |
| Mortality rate | | | | 0.0000139 | 0.0000138 | 0.0000135 | 0.0000243* |
| | | | | (0.268) | (0.277) | (0.286) | (0.053) |
| Vaccine rate (1st dose) | | | | | 0.0000155 | 0.000224 | 0.000291 |
| | | | | | (0.887) | (0.326) | (0.203) |
| Vaccine rate (2nd dose) | | | | | | -0.000266 | -0.000401 |
| | | | | | | (0.297) | (0.117) |
| Constant | -0.040**** | -0.373 | 0.094 | 0.119 | 0.124 | 0.104 | -0.134 |
| | (0.000) | (0.147) | (0.723) | (0.653) | (0.642) | (0.697) | (0.613) |
| $N$ | 2108 | 2100 | 2037 | 2037 | 2037 | 2037 | 2037 |
| $R^2$ | 0.021 | 0.321 | 0.376 | 0.377 | 0.377 | 0.377 | 0.370 |
| adj. $R^2$ | 0.021 | 0.318 | 0.371 | 0.371 | 0.371 | 0.371 | 0.364 |
| AIC | -5856.5 | -6607.7 | -6669.0 | -6668.3 | -6666.3 | -6665.4 | -6643.3 |

*Sources*: Ministry of Health, IBGE, TSE, Facebook Movement Range; authors' calculations.

\* $p < 0.10$

\*\* $p < 0.05$

\*\*\* $p < 0.01$

\*\*\*\* $p < 0.001$.

*Note*: OLS model. See Table 2 for the demographic and socioeconomic (1) as well as the dwelling and infrastructure (2) control variables.

namely lockdown and more broadly social distancing. As seen in Section 2, President Bolsonaro consistently derided this strategy, whether by opposing decisions made by sub-federal authorities (governors and mayors) or by means of a campaign of negative messages, particularly on social media.

The results of model (2) are presented in Table 3, which covers cumulative data up to the end of October 2021, following which mobility returned to its initial pre-pandemic level. The detailed table is available as S4 Table. For the sake of consistency and to facilitate comparison, we use the same specifications as for model (1) with the same set of independent variables. Only the mortality rate variable is added to some specifications (cols. 4–7). As the dependent variable consists of the difference in mobility compared with the pre-pandemic period, a positive coefficient corresponds to a lower reduction in mobility during the pandemic.

First, the unconditional correlation (col. 1) shows that, on average, the more pro-Bolsonaro the municipality, the more its population respected social distancing. However, once controls are introduced into the model, the coefficient is inversed. In all specifications, the '*Bolsonaro effect*' on mobility change, our main variable of interest, is at play in the expected direction. Reduction in population mobility was relatively lower in municipalities with more Bolsonaro voters. Whatever the specification, the coefficient is significant and stable (0.06). For example, a 10% increase in Bolsonaro supporters corresponds to around 0.6% more mobility. Given that mobility decreased by -6.4% on average from the start of the pandemic to October 2021 compared with the reference period (February 2020), we then have a gauge of the scope of the "*Bolsonaro effect*" on mobility. This result underpins the previous results regarding mortality.

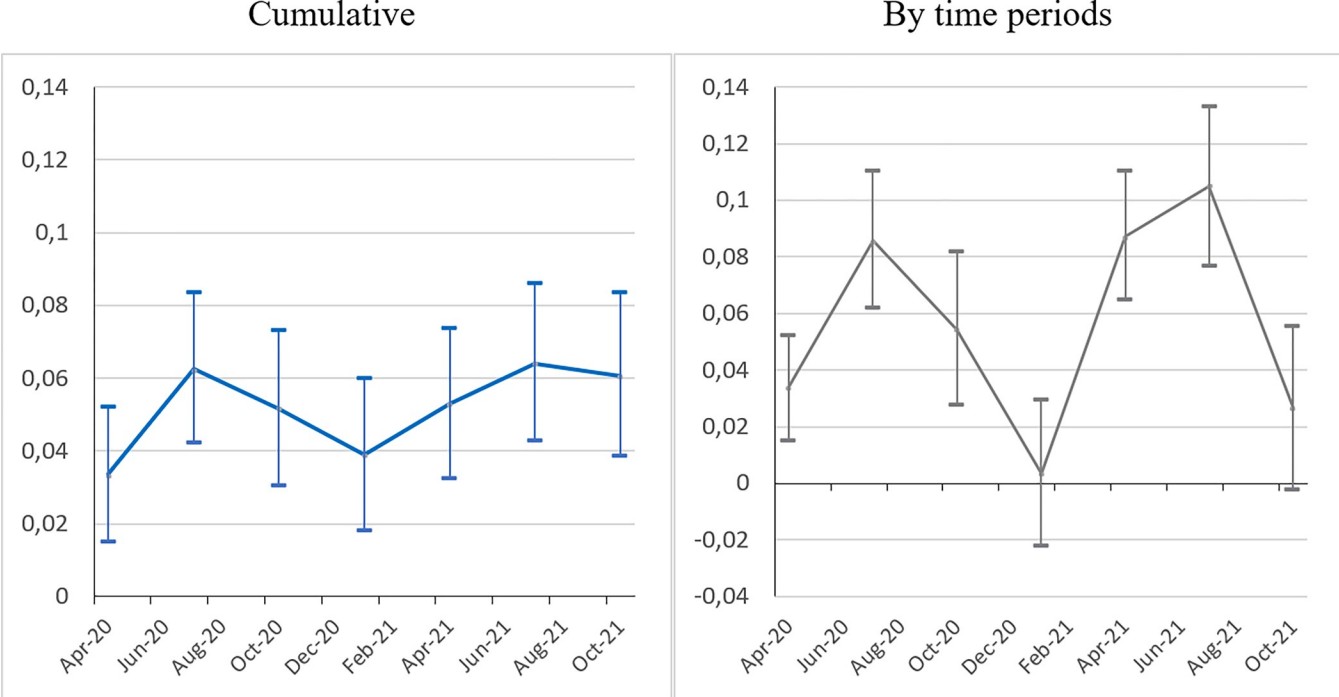

**Fig 4. The '*Bolsonaro effect*' on the change in mobility over time (2020–2021).** *Sources*: Ministry of ealth, IBGE, TSE, Facebook Movement Range; authors' calculations. *Note*: Model (2) (see Table 3, col.3). Confidence interval at $p < 0.05$.

It confirms that mobility is one of the transmission channels through which political factors have an impact on Covid-19 mortality.

Second, the leading structural factors driving changes in mobility behaviour and the inversion of the coefficient are age, race and education. Municipalities with older, white and better-educated people–three groups that voted more for Bolsonaro–stayed more at home. These groups correspond to those who were more able to work from home or did not work. Conversely, municipalities with a higher percentage of men reduced their mobility less, a result consistent with men generally adopting riskier behaviour by locking down less, while women were more inclined to stay at home due, among other things, to the sexual division of labour (care activities). Poor people appear to have respected social distancing less, probably because of a lack of resources and the need to go out to work in order to make a living. In addition, municipalities with higher population densities, more *favelas*, a higher concentration of dwellings or more commuters respected social distancing more. Finally, the mortality rate had a limited positive effect on mobility reduction, while vaccination rates had no significant effect.

As in the case of the mortality rate, the '*Bolsonaro effect*' on mobility changes is persistent over time. Fig 4 shows the temporal patterns of the Bolsonaro coefficient during the pandemic, for our preferred specification (without vaccination and mortality). Note that in terms of data availability, the number of municipalities with information on mobility is much higher at the start of the period (around 3,000 in 2020) and decreases gradually thereafter (down to 2,038 for the last quarter of 2021). Whether in cumulative data or by tranche (quarters), the '*Bolsonaro effect*' is significant and positive (equivalent of less reduction in mobility). By construction, the temporal profile for the cumulative data is smoothest, but both are quite similar. They follow the spread of the pandemic, at its height during the first and the second wave.

Two results are of note in addition to the '*Bolsonaro effect*'. First, in general, the coefficients of the controls hold (significance and sign) for all the periods, a result underpinning the main

conclusions drawn from the cumulative models. Second, the mortality rate has a positive impact on relative mobility in the five first quarters (from March 2020 to April 2021). Up to this date, municipalities with higher death rates due to Covid-19 tended to reduce their movements. Following this period, the coefficient remains negative (but not significant) between May and July 2021, and significantly positive during the last quarter (August-October 2021), which may explain why the impact of mortality on mobility is not significant for the entire period. Furthermore, during the active vaccination campaign periods (starting in February 2021), lower relative mobility is associated with a higher vaccination rate. This result suggests that the two strategies (stay at home and get vaccinated) were taken together, rejecting the hypothesis of the alternative of vaccination to be able to move around more at less risk.

## 4.3 Vaccination

The second channel through which the '*Bolsonaro effect*' can impact on the mortality rate is the main pharmaceutical measure of vaccination. As discussed in Section 1, President Bolsonaro rejected the vaccine. From the start, he actively obstructed the vaccination campaign and made many statements denigrating its efficiency.

As in the case of the mortality rate and changes in mobility, Table 4 presents the results of model (3) for doses 1 and 2 under three different specifications (no controls, cols. 1 and 4; sociodemographic characteristics, cols. 2 and 5; transmission factors, cols. 3 and 6) using cumulative data on vaccination through to December 2022. As in Table 2, to account for potential bias associated with unobserved, time-invariant state heterogeneity, the columns 3b and 6b present the results of the Poisson regression model with fixed effects at state level. However, for the same reasons as those given above for the mortality rate regressions, we will

**Table 4. Factors associated with vaccination rates (cumulative data: From January 2021 to December 2022).**

| | | | First dose | | Full schedule | | | |
|---|---|---|---|---|---|---|---|---|
| | (1) | (2) | (3a) | (3b) Poisson fe | (4) | (5) | (6a) | (6b) Poisson fe |
| **Vote for Bolsonaro** | 0.126**** | -0.143**** | -0.155**** | -0.230**** | 0.205**** | -0.231**** | -0.238**** | -0.315**** |
| **(1st round 2018)** | (0.000) | (0.000) | (0.000) | (0.001) | (0.000) | (0.000) | (0.000) | (0.000) |
| *Group of control variables* | | | | | | | | |
| Demographic and Socioeconomic (1) | | Yes | Yes | Yes | | Yes | Yes | Yes |
| Dwelling and | | | Yes | Yes | | | Yes | Yes |
| infrastructure (2) | | | | | | | | |
| Constant | 4.462**** | -2.196**** | -1.680**** | | 4.334**** | -3.608**** | -2.613**** | |
| | (0.000) | (0.000) | (0.000) | | (0.000) | (0.000) | (0.000) | |
| Lnalpha | -3.773**** | -4.793**** | -4.890**** | | -3.418**** | -4.522**** | -4.657**** | |
| | (0.000) | (0.000) | (0.000) | | (0.000) | (0.000) | (0.000) | |
| *N* | 5568 | 5340 | 5269 | 5268 | 5568 | 5340 | 5269 | 5268 |
| pseudo $R^2$ | 0.002 | 0.069 | 0.074 | | 0.004 | 0.081 | 0.088 | |
| *AIC* | 47149.2 | 42171.1 | 41409.1 | 41069.3 | 47619.2 | 42113.6 | 41252.6 | 40890.5 |

*Sources*: Ministry of Health, IBGE, TSE; authors' calculations.

* $p < 0.10$

** $p < 0.05$

*** $p < 0.01$

**** $p < 0.001$

*Note*: Negative Binomial (NB) model, except for the column 3b and 6b (Poisson State fixed effect).

See Table 2 for the demographic and socioeconomic (1) as well as the dwelling and infrastructure (2) control variables.

mainly focus on the results of the negative binomial model. The detailed table is available as S5 Table.

Here again, our evidence suggests a robust '*Bolsonaro effect*' on vaccination rates. The Bolsonaro coefficients are significant and negative for both the first and the second dose. Inhabitants of municipalities with a higher percentage of Bolsonaro voters posted a lower vaccination rate. The '*Bolsonaro effect*' is stronger for the full vaccination schedule, suggesting that Bolsonaro supporters may have complied less with international recommendations. For example, a 10% increase in Bolsonaro supporters corresponds to a 2% decrease in the vaccination rate (full schedule). This would suggest that 4 million fewer people nationwide had completed a full vaccination schedule by the end of 2022.

As in the case of the mobility models, vaccination rates were higher in municipalities with older, female, white and better-educated people. In addition, per capita GDP and life expectancy boosted vaccination rates. Interestingly enough, vaccination rates were also higher in municipalities with higher poverty rates and in rural areas. This factor may be due to the efficiency of the health system (SUS) providing access to free services throughout the country. However, this positive effect is offset by many other factors suggesting that poorer communities were worst covered. Such is the case with municipalities with more slums (*favelas*), more overcrowded dwellings and more informal jobs.

Fig 5 presents the results of the analysis of changes in Covid-19 vaccination rate patterns over time, by sub-periods. Once again, the '*Bolsonaro effect*' on the vaccination rate holds. For each quarter (except at the beginning for the quarter of May-July 2021) and for each setting (cumulative or by time period), the relationship is negative and significant for the 1st and 2nd dose, indicating lower vaccination rates in municipalities with a higher Bolsonaro vote. This result is supported by various individual surveys conducted over the period. A survey conducted in October 2020 reveals that Bolsonaro supporters are less likely to vaccinate than those who do not support him [59]. Nearly two years later (in August 2022), a survey sponsored by Sou Ciência confirmed this finding, this time not only in terms of intention but of actually being vaccinated [60]. The '*Bolsonaro effect*' on vaccination is all the more catastrophic in that Brazil is internationally recognized as a leader in vaccination campaigns. The longstanding National Immunization Program (NIP) has always been successful with a high level of support irrespective of individuals' political leanings. The strong adherence of the Brazilian population to immunization campaigns compared with other countries has been noted [46]. A paper on vaccine confidence in 67 countries confirms that the share of the population considering vaccination as important, safe and effective is generalized, placing Brazil in the top position worldwide [61]. Despite a decrease in initial resistance to vaccination by Bolsonaro supporters, as shown by polls on vaccination intentions (Poder360, various issues), the '*Bolsonaro effect*' on vaccination remains strong and significant through to the end of 2022.

In addition to investigating the '*Bolsonaro effect*' by time period, the data collected by the Ministry of Health can be used to disaggregate the vaccination rate by population categories. Three dimensions are captured: age, sex and skin colour (self-reported information). The data for this characteristic are not always reported on the vaccination forms. According to our estimates, this information is not available in about 18% of cases. As the population had the possibility of being vaccinated in different types of places (health centres, schools, ad-hoc vaccination centres, etc.) throughout the country, and as there are no systematic reasons for omitting this information, the missing values are in principle not likely to be a source of bias.

Fig 6 plots the coefficients obtained for the second dose for both cumulative data (up to May 2022) and by time period for the previous quarter (February-April 2022; last period for which a sufficient number of people were vaccinated) under the same specification as in Fig 5. The results are quite striking. First, the '*Bolsonaro effect*' is stronger for men than for women,

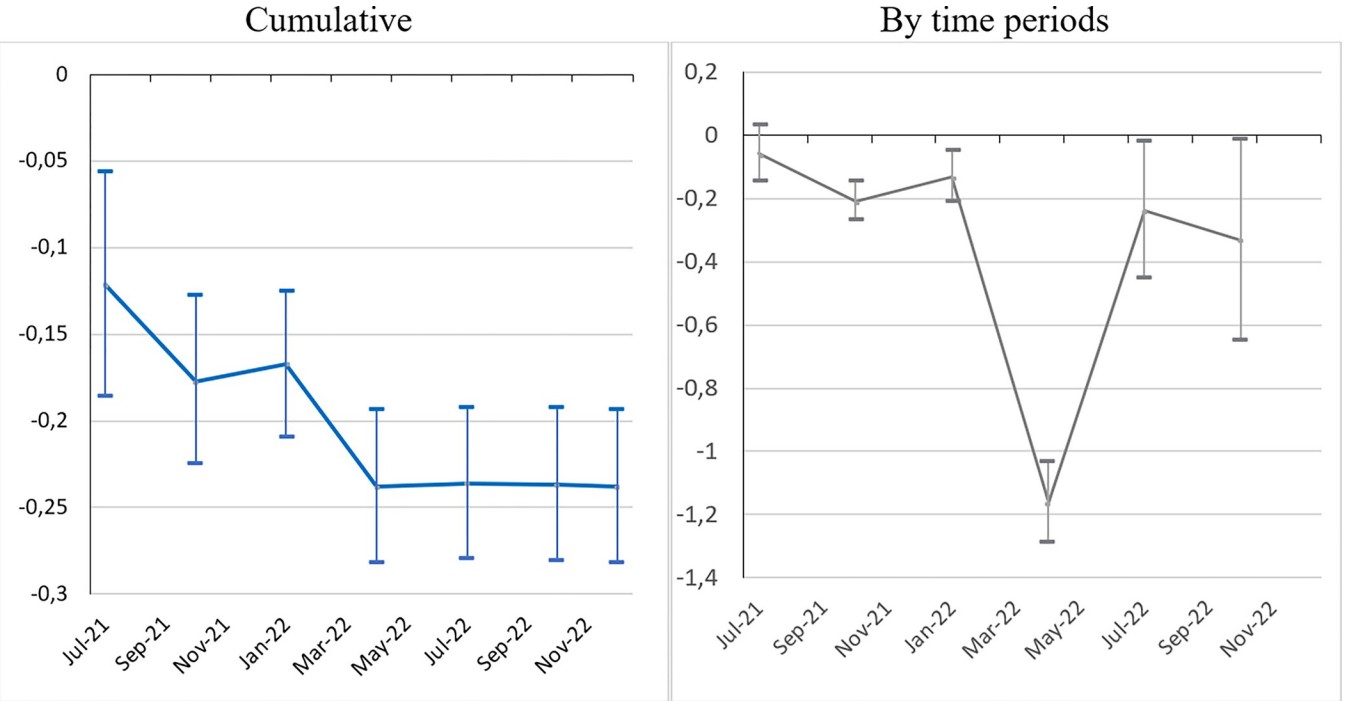

**Fig 5. The 'Bolsonaro effect' on the vaccination rate (2nd dose) over time (2021–2022).** *Sources*: Ministry of Health, IBGE, TSE; authors' calculations. *Note*: Model (3) (see Table 4, cols. 6a). Confidence interval at $p < 0.05$.

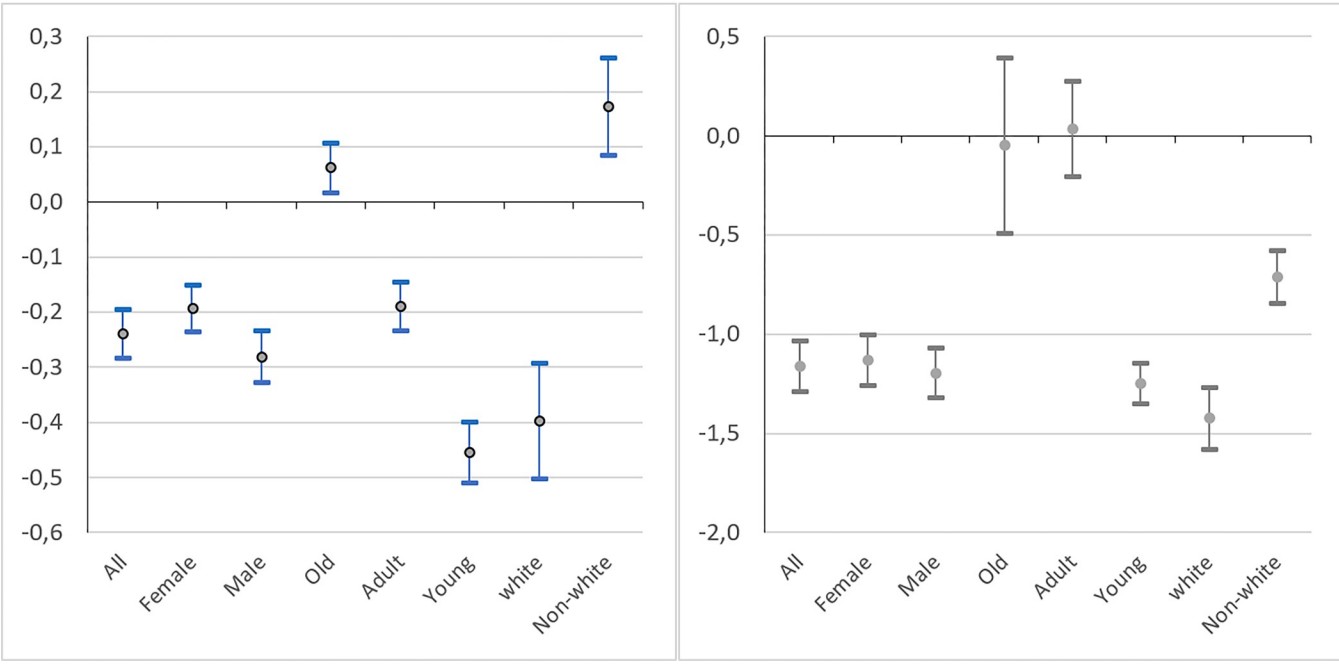

**Fig 6. The 'Bolsonaro effect' on the vaccination rate (2nd dose) by population categories.** *Sources*: Ministry of Health, IBGE, TSE; authors' calculations. *Note*: Model (3) (see Table 4, cols.6a). Confidence interval at $p < 0.05$.

but the difference is not significant. Second, there is a steep gradient by age bracket. The 'Bolsonaro effect' is systematically stronger for young people (under 30 years old) and decreases from there on. It is non-significant for both the older brackets (30 to 50 years old and over 50), albeit still negative for the intermediate age group and even slightly positive for the older bracket in the cumulative setting. Finally, in terms of race, the 'Bolsonaro effect' is at its height for white people. It is lowest for non-whites, albeit still negative in the cumulative data model and even positive for the Feb-Apr 2022 quarter. In other words, during this period, the vaccination rate for non-white people was higher in pro-Bolsonaro municipalities. To sum up, Bolsonaro's vaccination denialism had more of an effect on young and white people, while older and non-white people were less sensitive to (and sometimes even against) his negative messages.

Finally, we estimate our vaccination model (3) for other types of vaccination than for Covid-19 over different periods (S6 Table). As a placebo test, we find that there was no 'Bolsonaro effect' on the vaccination rate in 2017–2018 when Bolsonaro was not yet in power. In 2019, during the first year of his presidential term of office, but before the pandemic, there was still found to be no 'Bolsonaro effect'. However, the 'Bolsonaro effect' on other vaccination rates appears at significant levels in 2020, 2021 and 2022, increasing over time. Hence, not only did the aggregate non-Covid-19 vaccination rate drop during the Covid-19 pandemic (according to the Datasus, the indicator for the global vaccination coverage (for about 20 vaccines) was 77% in 2018, 73% in 2019, 68% in 2020, 61% in 2021 and 66% in 2022), but the decrease was greater in Bolsonarist municipalities. So anti-vaccine rhetoric associated with Covid-19 denialism appears to have been effective not only with respect to Covid-19 vaccination, but also with respect to traditional vaccination campaigns. One plausible interpretation is that Bolsonaro's rhetoric against Covid-19 vaccination created confusion among the population, generating a negative spillover effect on vaccination in general. Overall, the socioeconomic gradients are much less significant and pronounced, and the goodness of fit of the models is lesser for other vaccines than for Covid-19 vaccination, showing that in 'normal circumstances', the vaccination process is more under control and better distributed among social groups. Finally, above and beyond Covid-19, Bolsonaro's propaganda is having long-term effects that have already started to emerge, endangering the National Immunization Program acclaimed as an internationally recognized historical success [62, 63].

## 4.4 Robustness checks and extensions

In order to consolidate our results and take our study further, we conduct a set of robustness checks and test additional hypotheses. First, we estimate our models using excess mortality instead of the official data on Covid-19 fatalities. Two indicators of excess mortality are computed: total excess mortality and non-violent excess mortality. Our preferred benchmark to smooth the mortality rates prior to Covid-19 is the average of the previous three years (2017–2019). However, our results are robust to other reference periods (2017, 2018 and 2019). Bear in mind that it is not clear in the case of Brazil that excess mortality is a better indicator of the real Covid-19 mortality rate. First of all, as mentioned in Section 2, particular care was taken to record Covid-19 fatalities. Secondly, our indicator of excess mortality depends on the reference period chosen and Covid-19 is not the only reason for excess mortality, even when accounting for non-violent deaths only. Moreover, mortality decreased for some causes during the pandemic (traffic accidents, for instance). Mortality data are available through to August 2022. They are definitive for 2020 and provisional thereafter as they require further fastidious cleaning. Our estimates of excess mortality are plausible: 890,000 deaths vs 680,000 for Covid-19 deaths in August 2022. Our estimations confirm that the 'Bolsonaro effect' on the mortality rate is robust in all specifications and in the same region as with the official data (S7 Table).

The '*Bolsonaro effect*' is also slightly higher in the case of non-violent excess mortality, a better proxy for mortality due to Covid-19. Furthermore, the control coefficients are also robust over time and similar to those obtained with the official data.

Second, we test the impact of our variable of interest based on the second round of the 2018 presidential election. The '*Bolsonaro effect*' is still significant in both the cumulative model and for each period (S8 Table). However, the coefficients are smaller, suggesting that mortality was induced more by true Bolsonaro supporters. Indeed, supporters of President Bolsonaro knew significantly less about Covid-19 and coronavirus disease [21]. They would therefore be expected to be more inclined to adopt risky behaviour, as shown by their lower level of observance of social distancing and vaccination [28]. To investigate this point further, we compute a proxy for Bolsonaro voters in the second round as the difference in Bolsonaro voters between the second and first rounds. Estimation of the mortality rate (model 1) finds significant negative coefficients for this variable (except for the quarters when the spread of the virus slowed). Therefore, leaving aside these particular periods, the results provide suggestive evidence that those who voted for Bolsonaro in the second round were reluctant to accept Bolsonaro's anti-vaccination rhetoric, much in line with their socio-political profile (centrists and right-wing leaning; the so called '*terceira via*'–third way–between the left wing and the extreme right). This hypothesis is confirmed by our vaccination and mobility equations. Vaccination spread more in municipalities where Bolsonaro got more additional voters.

Taking our analysis further, we estimate the influence of Haddad's vote on mortality, mobility and vaccination. Haddad was the representative for the Workers' Party (PT), former President Lula's party, and Bolsonaro's main opponent in the 2018 election. He qualified for the second round. Again, the more municipalities voted for Haddad (first round), the more they were vaccinated and stayed at home and the fewer Covid-19 fatalities were recorded. This result is consistent with the study [20] which found, in an experimental design, that Bolsonaro voters were more optimistic about the health risks and job insecurity associated with Covid-19 at the start of the pandemic compared with those who voted for Haddad in the second round of the 2018 election. To sum up, the negative impact of Bolsonaro's denialism on mortality and mobility, while the other two political forces (centre and right on one side and left wing on the other) played a countervailing role, is confirmed, including for vaccination on the left.

Third, so far we have identified the '*Bolsonaro effect*' from the results of the 2018 presidential election. However, a new election was held in October 2022, when President Bolsonaro ran again. He qualified for the second round of the election with 43.2% of the vote as opposed to 48.4% for his opponent, former President Lula. Bolsonaro was eventually beaten, winning 49.1% of the vote in a tight ballot. Compared to the 2018 election, the vote for Bolsonaro fell by 3% in the 1st round. Moreover, the strong correlation between the two elections (0.97) shows that Bolsonaro enjoyed sound, enduring grassroots support over time. Yet quite significant variations also appear to be at work.

We therefore renew our estimates taking the results of the 2022 election. Econometrically speaking, it is not clear whether this dataset provides a better way to estimate the '*Bolsonaro effect*'. On the one hand, it has the advantage of better reflecting the state of current opinion, as there is no guarantee that the 2018 Bolsonarist municipalities were still such at the start of the pandemic in early 2020. Furthermore, the 2018 election was marked by very strong rejection of the PT, which drove part of the electorate (from the right and centre in particular) to vote for Bolsonaro in the first round. This was much less the case in 2022. On the other hand, the estimates are directly subject to the risk of reverse causality bias: President Bolsonaro's handling of the pandemic may have impacted the results of the 2022 election. This is why we consider these estimates based on 2022 data purely as a robustness test of the results obtained so far: Bolsonaro had a negative effect on the number of Covid-19 deaths, and this effect

channelled through less respect for social distancing and vaccination. The test is conclusive. All the models confirm the presence of a '*Bolsonaro effect*' on the three variables of interest. Moreover, the size of the effect is greater in most cases than with the 2018 data. Note that this is a conservative estimate of the "true" '*Bolsonaro effect*', since the potential reverse causality works in the opposite direction: a lower propensity to vote for him in the municipalities where there were more deaths. For reasons of space, we reproduce here only the models covering the first round of the election under the standard specification (S9 Table).

Fourth, in order to further identify Bolsonaro's own responsibility, we seek to purge the '*Bolsonaro effect*' from possible ideological interference, which could generate omitted variable biases. Two additional control variables are introduced into the models. First, we include the 2014 presidential election vote for the right-wing candidate, Aécio Neves, as an indicator of a conservative and potentially anti-science ideological ground pre-existing the emergence of Bolsonaro. Second, and in the same vein, the 2017–2018 general vaccination rate prior to Bolsonaro's arrival in power and the outbreak of the pandemic seeks to capture the same phenomenon more directly. It is important to note that this indicator measures the strength of the anti-vaccine movement, but that it is also correlated with the population's general health, including the presence of comorbidities. Indeed, it is likely that the municipalities with the highest rates of vaccination are also those with the most vulnerable populations.

In both cases, taking cumulative data for the entire period, the '*Bolsonaro effect*' persists and the coefficients remain of the same order of magnitude as those observed in the baseline models, whether for mortality, social distancing or vaccination. The coefficients associated with the 2014 Neves vote are non-significant in the mortality and mobility equation, and positive in the vaccination equation. However, the 2017–2018 vaccination rate coefficients are positive in the mortality equation, indicating that the comorbidities effect outweighs the "antivax" effect. Moreover, the '*Bolsonaro effect*' is particularly strong as it persists throughout the vast majority of sub-periods. Such is the case with the twelve successive quarters in cumulative data for the mortality rate. And although the results are less conclusive for mobility (with more non-significant coefficients), we still find the expected significant and negative sign for vaccination. To sum up, the empirical evidence suggests that the mechanism highlighted in this paper, embodied by a '*Bolsonaro effect*' on mortality, mobility and vaccination, is resistant to the inclusion of control variables related to preferences and values beyond support for President Bolsonaro, as well as to the prevalence of comorbidities in the population (S10 Table).

## 5. Conclusion

Brazil counts among the countries the hardest hit by the Covid-19 pandemic, which is all the more surprising given the country's reputation for the universal scope of its public health system (the SUS), its experience handling infectious diseases (Dengue, chikungunya, zica, etc.) and its well-developed National Immunization Program (NIP). Our results show the extent to which the statements and behaviour of a central figure in the public arena, here the President of the Republic, are likely to have a significant impact on the health and life of the population and jeopardize the assets and strengths of a health system built over time. To our knowledge, this paper presents the most comprehensive investigation of Bolsonaro's influence in the spread of the pandemic from two angles: considering both mortality due to Covid-19 and its two main transmission channels (social distancing and vaccination); and exploring the full pandemic cycle (2020–2022) and its dynamics over time.

The results for the '*Bolsonaro effect*' confirm the hypothesis that voter political orientation is related to the Covid-19 mortality rate and this relationship persisted over time. Furthermore, the results regarding the estimation of mobility also suggest that political stance tends to

influence compliance with social distancing measures, especially at the time when the pandemic reached its most critical levels in Brazil. With respect to vaccination, the '*Bolsonaro effect*', which does not appear clearly at the start of the vaccination process, is also subsequently confirmed. Thus, all other things being equal, the pro-Bolsonaro municipalities were proportionally less vaccinated. These results for mobility and vaccination highlight two mechanisms through which Bolsonaro supporters were relatively more affected by the virus, and probably contaminated their neighbours, irrespective of their political leanings. Various robustness tests were conducted to ensure that data reliability or endogeneity issues are limited.

Although the nature of the phenomenon analysed, with observational data, does not allow a causal relationship to be clearly identified, our specifications, which consider a large number of characteristics of municipalities, highlight significant and informative conditional correlations between political factors and the Covid-19 mortality rate. Our results are consistent with the hypothesis that most of Bolsonaro's supporters shared his denialist stance, which led them to adopt negligent attitudes in terms of social distancing or the use of masks, resulting in a higher level of mortality in the municipalities where they are numerous. These findings are in line with the empirical literature showing that countries with populist governments were worse at handling the pandemic crisis, with higher Covid mortality rates. Therefore, it points to the importance of the debate on how to design and guarantee consolidated public policies and solid institutions with a capillary effect on actions and information, reducing the vulnerability associated with populist political cycles.

Finally, looking forward to the future, one of the most worrying results is the observation of a '*Bolsonaro effect*' on general vaccination coverage. Over and above the negative impact on Covid-19 vaccination, our estimates support the hypothesis that Bolsonaro denialism impacted on compliance with the general vaccination schedule. This latter finding would need to be investigated further to see if it persists over time. A study would then be required to more specifically examine the profile of the population who became reluctant to vaccinate and the type of policies that might be put in place to remedy the situation.

## Supporting information

**S1 Table. Pandemic denialist statements by President Bolsonaro.** Sources: Various media; Authors' compilation.
(DOCX)

**S2 Table. Data sources and variables.** *Note*: *All variables are considered at municipal level.* *
adjusted at state level considering PNAD-C 2019.
(DOCX)

**S3 Table. Factors associated with the Covid-19 mortality rate—Detailed results (cumulative data: From February 2020 to December 2022).** *Sources*: Ministry of Health, IBGE, TSE; authors' calculations. * $p < 0.10$, ** $p < 0.05$, *** $p < 0.01$, **** $p < 0.001$. *Note*: Negative Binomial (NB) model, except for the last column (Poisson State fixed effect).
(DOCX)

**S4 Table. Factors associated with the change in mobility–Detailed results (cumulative data: From February 2020 to October 2021).** *Sources*: Ministry of Health, IBGE, TSE, Facebook Movement Range; authors' calculations. * $p < 0.10$, ** $p < 0.05$, *** $p < 0.01$, ****
$p < 0.001$. *Note*: OLS model.
(DOCX)

**S5 Table. Factors associated with vaccination rates–detailed results (cumulative data: From January 2021 to December 2022).** *Sources*: Ministry of Health, IBGE, TSE; authors' calculations. * $p < 0.10$, ** $p < 0.05$, *** $p < 0.01$, **** $p < 0.001$. *Note*: Negative Binomial (NB) model, except for the column 3b and 6b (Poisson State fixed effect).
(DOCX)

**S6 Table. Factors associated with Covid-19 and other disease vaccination rate (2017–2022).** *Sources*: Ministry of Health, IBGE, TSE; authors' calculations. * $p < 0.10$, ** $p < 0.05$, *** $p < 0.01$, **** $p < 0.001$. *Note*: Negative Binomial (NB) model.
(DOCX)

**S7 Table. Factors associated with excess mortality (cumulative data: Jan 2020 –Aug 2022).** *Sources*: Ministry of Health, IBGE, TSE; authors' calculations. *Note*: p-values in parentheses $p < 0.10$, ** $p < 0.05$, *** $p < 0.01$, **** $p < 0.001$. Negative Binomial model. The reference to compute excess mortality for each *municipio* is the average for the three-year period 2017–2019. For the first column, the Covid-19 mortality rate is the official figure (Ministry of Health).
(DOCX)

**S8 Table. Further exploration of the 'Bolsonaro effect' on the Covid-19 mortality rate (2018 election).** *Sources*: Ministry of Health, IBGE, TSE; authors' calculations. p-values in parentheses $p < 0.10$, ** $p < 0.05$, *** $p < 0.01$, **** $p < 0.001$. *Note*: Negative Binomial (NB) model. The control variables are always the same (those considered in Table 4) for each of the three specifications considered here (with only the percentage of votes for Bolsonaro in the first round; with only the percentage of votes for Bolsonaro in the second round; with the percentage of votes for Bolsonaro in the first round and the difference $2^{nd}$ -$1^{st}$ round).
(DOCX)

**S9 Table. The 'Bolsonaro effect' on Covid-19 mortality, social distancing and vaccination (2022 election) (cumulative data).** *Sources*: Ministry of Health, IBGE, TSE; authors' calculations. p-values in parentheses $p < 0.10$, ** $p < 0.05$, *** $p < 0.01$, **** $p < 0.001$. *Note*: Negative Binomial (NB) model. The control variables are always the same (those considered in Table 4) for each of the two specifications considered here (with only the percentage of votes for Bolsonaro in the first round in 2018; with only the percentage of votes for Bolsonaro in the first round in 2022.
(DOCX)

**S10 Table. The 'Bolsonaro effect' on Covid-19 mortality, mobility and vaccination rates controlling for preferences (cumulative data).** *Sources*: Ministry of Health, IBGE, TSE; authors' calculations. p-values in parentheses $p < 0.10$, ** $p < 0.05$, *** $p < 0.01$, **** $p < 0.001$. *Note*: Negative Binomial (NB) model. The control variables are always the same (those considered in Table 4; see also S9 Table) for each of the specifications considered.
(DOCX)

## Acknowledgments

We acknowledge seminar participants at the *Instituto de Economia* of the Federal University of Rio de Janeiro (IE/UFRJ) and IRD colleagues for comments on earlier drafts. We would like to thank also the reviewers, whose comments and suggestions helped us to improve the paper.

## Author Contributions

**Conceptualization:** Mireille Razafindrakoto, François Roubaud.

**Data curation:** Mireille Razafindrakoto, François Roubaud.

**Formal analysis:** Mireille Razafindrakoto, François Roubaud, Marta Reis Castilho, Valeria Pero, João Saboia.

**Methodology:** Mireille Razafindrakoto, François Roubaud.

**Validation:** João Saboia.

**Writing – original draft:** Mireille Razafindrakoto, François Roubaud, Marta Reis Castilho, Valeria Pero.

**Writing – review & editing:** Mireille Razafindrakoto, François Roubaud, Marta Reis Castilho, Valeria Pero, João Saboia.

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
