## [Decision Letter · Decision Letter 0]

7 Sep 2023

PONE-D-23-20876Investigating the ‘Bolsonaro effect’ on the spread of the Covid-19 pandemic: an empirical analysis of observational data in BrazilPLOS ONE

Dear Dr. RAZAFINDRAKOTO,

Thank you for submitting your manuscript to PLOS ONE. After careful consideration, we feel that it has merit but does not fully meet PLOS ONE’s publication criteria as it currently stands. Therefore, we invite you to submit a revised version of the manuscript that addresses the points raised during the review process.

ACADEMIC EDITOR: The manuscript presents updated results confirming that pro-Bolsonaro municipalities had higher fatality rates than other Brazilian municipalities. The article's theme is relevant, but there are already published articles showing more cases of COVID-19 and higher mortality in Brazilian municipalities where JB had more votes. Thus, for the present article to be considered for publication, the authors need to describe this previous research and be convincing about what the present article can add to the findings of these authors. Without this effort, the present article will be seen as "more of the same." The manuscript should also debate the centrality of STF decisions in combating the pandemic and Bolsonaro's attempts to prevent social distancing measures or mandatory vaccination. The author should also explain their methodological choices and the methods proposed to make causal inferences and be more comprehensive in presenting results. Finally, the conclusion should be refined to the broad literature about populism and democracy. 

We look forward to receiving your revised manuscript.

Kind regards,

Ivan Filipe de Almeida Lopes Fernandes, Ph.D.

Academic Editor

PLOS ONE

2. Please include a separate caption for each figure in your manuscript.

Additional Editor Comments:

The manuscript presents updated results confirming that pro-Bolsonaro municipalities had higher fatality rates than other Brazilian municipalities. The article's theme is relevant, but there are already published articles showing more cases of COVID-19 and higher mortality in Brazilian municipalities where JB had more votes. Thus, for the present article to be considered for publication, the authors need to describe this previous research and be convincing about what the present article can add to the findings of these authors. Without this effort, the present article will be seen as "more of the same." The manuscript should also debate the centrality of STF decisions in combating the pandemic and Bolsonaro's attempts to prevent social distancing measures or mandatory vaccination. The author should also explain their methodological choices and the methods proposed to make causal inferences and be more comprehensive in presenting results. Finally, the conclusion should also be refined to the broad literature about populism and democracy.

Reviewers' comments:

Reviewer's Responses to Questions

**Comments to the Author**

1. Is the manuscript technically sound, and do the data support the conclusions?

Reviewer #1: Yes

Reviewer #2: Partly

2. Has the statistical analysis been performed appropriately and rigorously? 

Reviewer #1: I Don't Know

Reviewer #2: N/A

3. Have the authors made all data underlying the findings in their manuscript fully available?

Reviewer #1: Yes

Reviewer #2: Yes

4. Is the manuscript presented in an intelligible fashion and written in standard English?

Reviewer #1: Yes

Reviewer #2: Yes

5. Review Comments to the Author

Reviewer #1: The paper is well written and has a clear and relevant research problem. The data used demonstrates that the main argument is well supported.

I am not able to assess the quality of the statistical tests presented, but would like to draw attention to issues that could be improved in the text.

First, it is important to emphasize the relevance of the paper not only for what it adds when compared to the existing literature. It failed to present the reader the relevance of the work to a broader discussion, such as the impact of populist and anti-science leaders for the development of evidence-based public policies and, specifically, for facing health crises, such as Covid-19.

Regarding the works already published on the subject, I missed one that had repercussions in Brazil, that of Fernandes and Fernandes (2022), "Populism and health. An evaluation of the effects of right-wing populism on the COVID-19 pandemic in Brazil"

There is a vast literature on the effects of populist leaders on democracy and its policies, and this debate should be mentioned in the paper's justification.

Second, the paper has not mentioned a set of existing works in the national literature that mention the centrality of STF decisions in combating the pandemic and Bolsonaro's attempts to prevent social distancing measures or mandatory vaccination. Studies such as Fernandes and Ouverney (2022) and Oliveira and Madeira (2021), among others, have presented this debate.

Finally, the conclusions were very much centered on the empirical findings, but failed to rescue the broader debate in which it is embedded, including the one mentioned above - the impact of populist leaders on the conduct of evidence-based public policy.

Reviewer #2: The manuscript titled "Investigating the 'Bolsonaro effect' on the spread of the COVID-19 pandemic: an empirical analysis of observational data in Brazil" presents updated results confirming that pro-Bolsonaro municipalities had higher fatality rates compared to other Brazilian municipalities.

In terms of the literature review, I recommend more extensive engagement with the article "Populism and health: An evaluation of the effects of right-wing populism on the COVID-19 pandemic in Brazil," published in PlosOne.

To address potential bias associated with unobserved factors related to time and geography, it is advisable to include day and fixed-effects in your analysis. I discourage the use of the Negative Binomial (NB) model with fixed effects (FEs). Instead, it would be preferable to use either Linear Probability Models (LPM) or Poisson regression with fixed effects. A valuable reference for this approach is Wooldridge, J. M.'s 1999 paper, "Distribution-Free Estimation of Some Nonlinear Panel Data Models" in the Journal of Econometrics (Vol. 90, pp. 77–97).

Tables should offer comprehensive details regarding variable measurements and technical information, reducing the need for readers to refer back to the article for clarification.

Furthermore, I recommend excluding the last columns of Tables 1 and 2 if you do not intend to interpret the controls. This can streamline the presentation of results and emphasize the key findings.

For a scientific journal, it is advisable to consider shortening your article to enhance clarity and improve communication with your readers.

Lastly, given the causal nature of your research, it is imperative to clarify the causal interpretation of the main coefficient and provide robust justification. Explain why this coefficient can be interpreted causally and offer supporting evidence or reasoning.

6. PLOS authors have the option to publish the peer review history of their article (what does this mean?). If published, this will include your full peer review and any attached files.

Reviewer #1: No

Reviewer #2: **Yes: **Gabriel Cepaluni

---

## [Author Response · Author response to Decision Letter 0]

15 Nov 2023

Please see the file "Response to the reviewers". Many thanks

---

## [Decision Letter · Decision Letter 1]

19 Jan 2024

PONE-D-23-20876R1Investigating the ‘Bolsonaro effect’ on the spread of the Covid-19 pandemic: an empirical analysis of observational data in BrazilPLOS ONE

Dear Dr. RAZAFINDRAKOTO,

Thank you for submitting your manuscript to PLOS ONE. After careful consideration, we feel that it has merit but does not fully meet PLOS ONE’s publication criteria as it currently stands. Therefore, we invite you to submit a revised version of the manuscript that addresses the points raised during the review process.

The manuscript presents updated results confirming that pro-Bolsonaro municipalities had higher fatality rates than other Brazilian municipalities. The paper's theme is relevant, and interesting and fundamental changes were made to the manuscript that made the study more suitable for a journal such as Plos One. However, some minor adjustments are still necessary for the manuscript to be published, as indicated by one of the reviewers, such as the issues of ecological fallacy, the choice and reasoning of controls, and omitted variabel bias, which need to be answered in the paper or justified to the reviewer.

We look forward to receiving your revised manuscript.

Kind regards,

Ivan Filipe de Almeida Lopes Fernandes, Ph.D.

Academic Editor

PLOS ONE

Journal Requirements:

Additional Editor Comments:

The manuscript presents updated results confirming that pro-Bolsonaro municipalities had higher fatality rates than other Brazilian municipalities. The paper's theme is relevant, and interesting and fundamental changes were made to the manuscript that made the study more suitable for a journal such as Plos One. However, some minor adjustments are still necessary for the manuscript to be published, as indicated by one of the reviewers, such as the issues of ecological fallacy, the choice and reasoning of controls, and omitted variabel bias, which need to be answered in the paper or justified to the reviewer.

Reviewers' comments:

Reviewer's Responses to Questions

**Comments to the Author**

1. If the authors have adequately addressed your comments raised in a previous round of review and you feel that this manuscript is now acceptable for publication, you may indicate that here to bypass the “Comments to the Author” section, enter your conflict of interest statement in the “Confidential to Editor” section, and submit your "Accept" recommendation.

Reviewer #1: All comments have been addressed

Reviewer #2: (No Response)

2. Is the manuscript technically sound, and do the data support the conclusions?

Reviewer #1: Yes

Reviewer #2: Yes

3. Has the statistical analysis been performed appropriately and rigorously? 

Reviewer #1: Yes

Reviewer #2: I Don't Know

4. Have the authors made all data underlying the findings in their manuscript fully available?

Reviewer #1: Yes

Reviewer #2: Yes

5. Is the manuscript presented in an intelligible fashion and written in standard English?

Reviewer #1: Yes

Reviewer #2: Yes

6. Review Comments to the Author

Reviewer #1: The article has been carefully revised, taking into account the comments of the two reviews. The revision has helped improve the article and the result is now suitable for publication.

Reviewer #2: Thank you for giving me the opportunity to review the revised version of “Investigating the 'Bolsonaro effect' on the spread of the Covid-19 pandemic.”

I find the article promising, but it still requires several improvements before it can be published.

Firstly, I noticed that the revised version is missing the figures and the appendix. A letter from the authors explaining the revisions would have been highly beneficial for me.

On page 12, the claim that “a significant effect at the municipal level guarantees significance in terms of individual probabilities” is not accurate. This overlooks the ecological fallacy, an important statistical concept. Significant effects at a municipal level do not necessarily translate to similar relationships at an individual level, as aggregate data can mask significant variations. Thus, it's crucial to analyze and interpret data at the appropriate level to avoid incorrect inferences.

Still on page 12, the authors’ dismissal of the potential influence of omitted variables correlated with political factors is unconvincing. Many factors could impact the relationship between support for Bolsonaro and the spread of Covid-19. For example, pre-existing characteristics of the municipalities, such as distrust in science or social behavior patterns, might influence this relationship. I believe the authors should focus on acknowledging the limitations of their research design rather than defending it weakly.

I suggest that on page 13, instead of using bullet points, a chronological figure would be more effective for clarity.

The authors do not disclose the controls in their regressions, which makes the analysis on pages 18 and 23 challenging to interpret. They should spend more time explaining their choice of controls, referencing the work of Carlos Cinelli, Andrew Forney, and Judea Pearl from 2022 on differentiating between “good” and “bad” controls.

Since the authors frequently praise Brazil’s immunization system, I believe it is essential for them to present more substantial evidence and cite relevant literature to support these claims.

For minor comments:

- In the first paragraph, I suggest using an active voice and clearly stating that Brazil had the second-highest absolute number of Covid-19 deaths.

- The term 'Covid-19' should be used consistently throughout the text.

7. PLOS authors have the option to publish the peer review history of their article (what does this mean?). If published, this will include your full peer review and any attached files.

Reviewer #1: No

Reviewer #2: **Yes: **Gabriel Cepaluni

---

## [Author Response · Author response to Decision Letter 1]

27 Jan 2024

see the file "Response to reviewers" 

Detailed answers to Reviewers

We would like to thank the editor and reviewers for their positive assessment of our article and their valuable comments. The suggestions helped us to improve the quality of the document. We have taken them all into consideration in our revision. We believe that the manuscript has been significantly improved and we hope that this revised version meets the journal's requirements for publication. In any case, we remain at your disposal if you have any further questions.

Please find below our responses (in black) regarding the points you raised (in blue). To facilitate the iden-tification of our modifications, they appear in the revised manuscript (revision 2) as ‘track changes’.

Reviewer #1: The article has been carefully revised, taking into account the comments of the two reviews. The revision has helped improve the article and the result is now suitable for publication.

We would like to thank reviewer #1 for this comment in recognition of the efforts made to carefully revise and improve the manuscript (responding to and considering all the comments made in the previous review).

Reviewer #2: Thank you for giving me the opportunity to review the revised version of “Investigating the 'Bolsonaro effect' on the spread of the Covid-19 pandemic.”

Thank you for giving me the opportunity to review the revised version of “Investigating the 'Bolsonaro effect' on the spread of the Covid-19 pandemic.”

I find the article promising, but it still requires several improvements before it can be published.

Firstly, I noticed that the revised version is missing the figures and the appendix. A letter from the authors explaining the revisions would have been highly beneficial for me.

First, we must stress that we have prepared and sent an extensive Answer to the Reviewers, with special attention to Reviewer#2 comments, who carefully reviewed our paper in the previous round (revision 1). We deeply regret the fact that reviewer 2 did not receive our detailed responses which consider all the comments (as pointed out by reviewer#1; see above). 

The figures and the appendix were also sent with the revised manuscript and the response to the reviewers.

In his new review by Reviewer#2, some comments were already addressed in our first “Answer to the Reviewers”, while other are new comments, not stressed earlier. Thus, we will send our initial (and augmented) answers to the old comments (taking advantage to re-send the full Answer) and elaborate on the new queries. 

Reviewer #2:

On page 12, the claim that “a significant effect at the municipal level guarantees significance in terms of individual probabilities” is not accurate. This overlooks the ecological fallacy, an important statistical concept. Significant effects at a municipal level do not necessarily translate to similar relationships at an individual level, as aggregate data can mask significant variations. Thus, it's crucial to analyze and interpret data at the appropriate level to avoid incorrect inferences.

First, we make it clear in the paper that we do not interpret our results in terms of individual probabilities. Our approach and the analysis is carried out at the municipal level.

Regarding the comment on the inaccuracy of this specific sentence “a significant effect at the municipal level guarantees significance in terms of individual probabilities”, we disagree with this point. Let us suppose there is no individual effect (correlation, conditional or unconditional) at the individual level (take for instance by gender). Necessarily, there should not be any effect at the aggregate (here municipal) level (P => Q). In consequence, as (No Q => No P), an effect at the aggregate (municipal) level implies necessarily the existence of an effect at the individual level. What is not known is the sign of the effect, although the probability the correlation changes its sign is limited, but it is still possible. 

This argument is explicitly developed in:

McLaren, J. (2021). Racial disparity in COVID-19 death: Seeking economic roots with Census data. The B.E. Journal of Economic Analysis & Policy, 21(3), Doi: 10.1515/bejeap-2020-0371, p.899:

‘The data used are US county aggregates of COVID-19 mortality over time, matched with socioeconomic and demographic data by country. The mortality data are not broken down by race, but in principle one can still test for and measure racial disparity by looking for correlation between the size of a county’s minority population share and the county’s mortality rate. If there were no difference in mortality between a minority group and the rest of the population, the size of the minority share would have no effect on the county’s mortality rate, so this correlation provides indirect evidence of a disparity in mortality rates. The exercise follows two steps: first, confirm the existence of a racial disparity by using a regression to show that minority population shares are correlated with mortality rates. Second, control for a range of county-level socio-economic factors to see if the racial disparity weakens or disappears. If it does, then that stands as evidence that the socioeconomic factor is part of the reason for the racial mortality discrepancy”

In our paper, we have specified that :

“However, this type of approach has its limitations and the results should be interpreted with due caution. First, analysis by municipalities cannot be interpreted in terms of individual risks. However, we know that a significant effect at municipal level also guarantees significance in terms of individual probabilities. We can assume, then, that the individual and municipal approaches generally converge in terms of signs (McLaren, 2021). Otherwise, inverse mechanisms would need to be explained. ”

Reviewer #2:

Still on page 12, the authors’ dismissal of the potential influence of omitted variables correlated with political factors is unconvincing. Many factors could impact the relationship between support for Bolsonaro and the spread of Covid-19. For example, pre-existing characteristics of the municipalities, such as distrust in science or social behavior patterns, might influence this relationship. I believe the authors should focus on acknowledging the limitations of their research design rather than defending it weakly.

New comment. The reviewer is right: the omitted variable bias cannot be excluded. But this is the case of all analysis on observational data (unless credible empirical strategies – like IV or RDD – can be implemented; which is not our case). However, in the specific example of political factors mentioned by the reviewer (distrust in science, social behaviour patterns), we do test this potential bias, at least for vaccination. As the “Bolsonaro effect” is not significant on other vaccines in 2019 (Bolsonaro was not in charge; placebo test), but only since 2020 on Covid-19 vaccination (and other vaccines), we can plausibly induce that we are capturing a real “Bolsonaro effect”, based on his specific attitudes and behaviours regarding the pandemic. Furthermore, logistic regressions based on an individual survey sponsored by Sou Ciência in 2022 show the “Bolsonaro effect” holds controlling for religion (evangelic, correlated with Bolsonaro support and anti-science attitudes) and other sociodemographic variables on vaccination, Covid-19 testing and masks (results available upon request to the authors).

In all case, we explicitly highlight the potential issue of omitted variables (see our first Answer to the Reviewers). 

In our paper, we have specified that : 

“the econometric models tested here can be used to estimate the correlations between the mortality rate and different factors, corrected for structural effects. However, as in most analyses of observational data, we identify correlations that do not necessarily imply causality. For example, it is quite plausible that restrictions were applied more strictly in municipalities where mortality rates were already higher. It is therefore hard to disentangle the actual impact of the restrictive measures. Finally, there is the issue of potentially omitted variables correlated with political factors. In this case, the correlation observed in our model might be influenced by these omitted variables. Yet it is hard to find convincing arguments to support this reservation. Considering the data on comorbidities most often cited as examples of omitted variables, it is not clear to what extent they might be correlated with political factors” (Section 3.1, Empirical approach); 

and “Although the nature of the phenomenon analysed, with observational data, does not allow a causal relationship to be clearly identified, our specifications, which consider a large number of characteristics of municipalities, highlight significant and informative conditional correlations between political factors and the Covid-19 mortality rate.” (Conclusion)

Reviewer #2:

I suggest that on page 13, instead of using bullet points, a chronological figure would be more effective for clarity.

We thank the reviewer for this suggestion, which makes the presentation clearer.

A figure presents the timeline of the Covid-19 pandemic instead of bullet points. 

Reviewer #2:

The authors do not disclose the controls in their regressions, which makes the analysis on pages 18 and 23 challenging to interpret. They should spend more time explaining their choice of controls, referencing the work of Carlos Cinelli, Andrew Forney, and Judea Pearl from 2022 on differentiating between “good” and “bad” controls.

First, we would like to point out that the complete tables with the controls are available in the appendix provided with the manuscript (table S2, S3, S4, etc.). We have been advised to simplify the presentation of the tables in the body of the text, bearing in mind that our main objective is to better identify the 'Bolsonaro effect' rather than to interpret all the controls per se.

But, we would like also to thank the reviewer for this suggestion to provide more explanation on the choice of control variables. We agree, and we think that quoting this reference (Cinelli et al., 2022) is highly relevant for identifying bad and good controls. We have therefore taken into consideration this suggestion. 

However, as we have already been asked to shorten the manuscript, we have essentially added explanations in cases where the choice of introducing controls was debatable and potentially problematic. 

In reference to Cinelli et al. (2022), we consider in our models a broad spectrum of controls: 1/variables with a potential direct or indirect effect on the mortality rate (neutral control, possibly good for precision) and above all, 2/ confounding variables which potentially might have influence on the mortality rate and the political factor (good controls aiming at blocking back-door paths).

We specified in the manuscript that “Among the control variables, the vaccination rate and mobility indicator are two transmission factors that could be considered. However, their inclusion in the model would potentially absorb at least part of the effect of our variable of interest (the political factor). This could lead to overcontrol, as our interest is in the total effect (Cinelli et al., 2022). In addition, it should be noted that the introduction of these two variables could engender a reverse causality problem in model 1. … In the case of the vaccination rate, when considering the 2nd dose of the vaccine, controlling for the 1st dose is a way to purge the reverse causality effect from the ratio of mortality to the number of vaccines. So, the vaccination rate remains in one of our specifications, but since it could be classified as a ‘bad control’ or a collider, our preferred specification also excludes the vaccination rate.” 

In the discussion, we acknowledge that the introduction of the vaccination rates as control variables could be questioned [Cinelli et al., 2022], but, we have decided to keep this step as a sensitivity test.

Reviewer #2:

Since the authors frequently praise Brazil’s immunization system, I believe it is essential for them to present more substantial evidence and cite relevant literature to support these claims.

We thank the reviewer for this recommendation. 

Even though we have already cited references highlighting the success of the vaccination pro-gramme in Brazil: 

46. Bernardeau-Serra, L., Nguyen-Huynh, A., Sponagel, L., Sernizon Guimarães, N., Teixeira de Aguiar, R.A., Soriano Marcolino, M. (2021), “The COVID-19 Vaccination Strategy in Brazil – A Case Study”, Epidemiolo-gy 202 (2): 338-359. 

60. Larson, H.J., de Figueiredo, A., Xiahong, Z., Schulz, W.S., Verger, P., Johnston, I.G., Cook, A.R., Jones N.S. (2016), “The State of Vaccine Confidence 2016: Global Insights Through a 67-Country Survey”, EbioMedi-cine, 12: 295-301. doi: 10.1016/j.ebiom.2016.08.042.

We provide here additional references which support the assertion that the National Immunization Program (NIP) is considered as one of the most successful immunization programs. We have in-cluded two of them in our revised manuscript.

Homma, A., Martins, R. d. M., Leal, M. d. L. F., Freire, M. d. S. & Couto, A. R. (2011), ‘Atualização em vacinas, imunizações e inovação tecnológica’, Ciência e Saúde Coletiva 16(2), 445–458.

In this study, researchers from Fiocruz claim that:

“The NIP is the most effective programme among emerging countries and is comparable to that of developed countries. Its enormous growth can be demonstrated with the following figures: in 2000, the budget was R$200 million; it rose to R$825 million in 2009, offering 26 different types of vac-cines....”

Domingues, C. M. A. S., Maranhão, A. G. K., Teixeira, A. M., Fantinato, F. F. S., & Domingues, R. A. S. (2020). 46 anos do Programa Nacional de Imunizações: uma história repleta de conquistas e desafios a serem superados. Cadernos De Saúde Pública, 36, e00222919. https://doi.org/10.1590/0102-311X00222919

Sato, A. P. S.. (2018). What is the importance of vaccine hesitancy in the drop of vaccination cov-erage in Brazil?. Revista De Saúde Pública, 52, 96. https://doi.org/10.11606/S1518-8787.2018052001199

Pércio J, Fernandes EG, Maciel EL, Lima NVT. 50 years of the Brazilian National Immunization Program and the Immunization Agenda 2030. Epidemiol Serv Saude. 2023 Dec 4;32(3):e20231009. doi: 10.1590/S2237-96222023000300001.

For minor comments:

Reviewer #2:

- In the first paragraph, I suggest using an active voice and clearly stating that Brazil had the second-highest absolute number of Covid-19 deaths.

Many thanks, we took into consideration this suggestion and made the changes in the manuscript.

Reviewer #2:

- The term 'Covid-19' should be used consistently throughout the text.

We are sorry but we do not understand this comment. We had our paper proofread by a native English speaker. On checking, we used the term Covid-19 consistently to refer to the Coronavirus disease 2019. In the text, we use the expression "the Covid-19 pandemic" on a few rare occasions when we consider it necessary to emphasise the fact that this is a pandemic.

---

## [Editor Report · Decision Letter 2]

2 Feb 2024

Investigating the ‘Bolsonaro effect’ on the spread of the Covid-19 pandemic: an empirical analysis of observational data in Brazil

PONE-D-23-20876R2

Dear Dr. Razafindrakoto,

We’re pleased to inform you that your manuscript has been judged scientifically suitable for publication and will be formally accepted for publication once it meets all outstanding technical requirements.

Kind regards,

Ivan Filipe de Almeida Lopes Fernandes, Ph.D.

Academic Editor

PLOS ONE
---

## [Editor Report · Acceptance letter]

16 Feb 2024

PONE-D-23-20876R2 

PLOS ONE

Dear Dr. Razafindrakoto, 

I'm pleased to inform you that your manuscript has been deemed suitable for publication in PLOS ONE. Congratulations! Your manuscript is now being handed over to our production team.

Kind regards, 

on behalf of

Dr. Ivan Filipe de Almeida Lopes Fernandes 

Academic Editor

PLOS ONE